

# Storyline Analytical Framework for Understanding Future Severe Low-Water Episodes and Their Consequences

Gabriel Rondeau-Genesse[1], Louis-Philippe Caron[1], Kristelle Audet[2], Laurent Da Silva[4], Daniel Tarte[3], Rachel Parent[3], Élise Comeau[5], and Dominic Matte[1,5]

[1]Ouranos inc., 550 Rue Sherbrooke W., West Tower, 19th floor, Montreal (QC), Canada, H3A 1B9
[2]Groupe AGÉCO, 25 Mozart Avenue East, office 310, Montreal (QC), Canada, H2S 1B1
[3]T[2] Environnement, 91 boul. Constable, McMasterville (QC), Canada, J3G 1M8
[4]Nada Conseils, Montreal (QC), Canada
[5]Université du Québec à Montréal (UQÀM), Montreal (QC), Canada

**Correspondence:** Gabriel Rondeau-Genesse (rondeau-genesse.gabriel@ouranos.ca)

**Abstract.**

The 2021 drought highlighted the vulnerability of Quebec's water resources and the potential for widespread consequences in a region that is generally perceived as having abundant water. This study uses a storyline approach to explore the plausible impacts of future drought conditions for an event similar to what occurred in 2021, but under two different warming scenarios corresponding to increases of 2°C and 3°C in global surface temperatures compared to preindustrial levels. The approach employs analogues derived from a large ensemble of regional climate simulations combined with simulations generated by a hydrological model to offer a comprehensive understanding of both climate and hydrological conditions during, and leading up to, these potential future events. This approach allowed for enhanced collaboration with water management experts and other stakeholders to project the possible impacts of climate change on serious water deficits in Quebec. Results indicate a further deterioration in river conditions, particularly under a +3°C global temperature rise. In the hardest-hit areas of the province under that scenario, future low-water levels persist for a month longer and river streamflows drop by an additional 50%, thus falling short of the threshold required to maintain the health of ecosystems for an extended period of time and suggesting significant impacts on ecosystems and human activities. This study also highlights the need for improved systematic data collection during meteorological and hydrological droughts in Quebec, particularly with respect to their impacts on human activities and ecosystems.

## 1 Introduction

The province of Quebec in Canada is generally perceived as a region with abundant water, since spring snowmelt recharges groundwater and ensures that most streams maintain water flow throughout the year, while preventing most multi-year droughts (Dubois et al., 2021; Assani, 2023). However, recent years have shown that the region is not impervious to low-water episodes. For example, Quebec's agricultural areas faced moderate to severe drought conditions in 2001, 2002 and for 8 of the 12 years between 2010 and 2021, each event resulting in financial impacts of tens of millions of dollars (Nand and Qi, 2023). A drought




in 2012 also severely affected the boreal forest (Houle et al., 2016), while, more recently, the extreme 2023 wildfire season in northern Quebec followed one of the driest and warmest seasons on record (Boulanger et al., 2024).

Analyzing hydrological droughts in colder regions can be challenging due to the interplay of rainfall, snow and temperatures.

The main factors influencing low water levels during summer differ according to the specific characteristics and geographical location of the basin. In Europe, for example, 50% of droughts are due to a rainfall deficit, while the remaining 50% (and often the worst) are caused by a series of other phenomena, either in isolation or in combination, such as warm snow seasons that lead to a lack of recharge in spring, rainfall deficits that extend into snowfall deficits, or delayed snow or glacier melt due to cold conditions (Van Loon and Van Lanen, 2012; Van Loon et al., 2015). Even when a single type of drought is examined, such

as the lack of snowmelt-induced aquifer recharge, the dominant factors causing them can vary depending on the location. In Austria, for example, snowmelt droughts are primarily caused by warm and dry winter and spring conditions, while in Norway, this type of drought is mainly controlled by winter precipitation alone (Van Loon et al., 2015). In southern Quebec, the total amount of rainfall after the melting of the snowpack is generally the main predictor of the intensity of low-water conditions in summer, followed by the total amount of evapotranspiration (Kinnard et al., 2022). However, in certain watersheds, other

variables, such as the maximum snow water equivalent ($SWE_{max}$), the date of onset of melting and the duration of snowmelt can explain water scarcity to the same extent as evapotranspiration (Kinnard et al., 2022). This is because during low-water periods in summer, a considerable portion of streamflow is derived from groundwater flow, and a prolonged and intense spring thaw helps recharge aquifers.

A significant rise in temperature is anticipated in eastern Canada due to human-induced climate change, which is very likely

to shorten the duration of snow cover and could decrease $SWE_{max}$ by 2.5 to 7.5% per decade through 2050 in southern Quebec, relative to 1981-2015 (Mudryk et al., 2018; Derksen et al., 2019; IPCC, 2023). Increasing temperatures will result in higher total annual precipitation, but this increase will mainly be constrained to spring and winter, with minimal changes in summer and fall (Alberti-Dufort et al., 2022a; IPCC, 2023). This rise in annual precipitation is expected to enhance groundwater recharge on average. However, this will be accompanied by more pronounced seasonal variations, leading to increased recharge rates in

winter and decreased rates in summer (Dubois et al., 2021, 2022). Coupled with higher evapotranspiration rates due to rising temperatures, summer water availability in southern Quebec is therefore projected to decline in the future (Bonsal et al., 2019; Alberti-Dufort et al., 2022a; IPCC, 2023). Consequently, the latest studies suggest that a reduction in average summer flow and an increase in the number of extreme low-flow events are likely for watersheds in Quebec (Aygün et al., 2019; Bonsal et al., 2019; Thackeray et al., 2019; Charron et al., 2021; Alberti-Dufort et al., 2022b; Kinnard et al., 2022).

A decrease in available water during the summer months, coupled with a higher demand due to population growth and anthropogenic climate change in various sectors (such as agriculture, industry and municipal services), could exacerbate scarcity, decrease water quality and give rise to conflicts over water usage (Mehdi et al., 2015; Furber et al., 2016; Wheaton and Kulshreshtha, 2017; Osman, 2018; Chilima et al., 2021). Despite some advances on this issue in recent years, the implications of acute water shortages in Quebec are not yet well documented or understood, making it difficult to anticipate their consequences,

especially for cases of more extreme drought conditions (Charron et al., 2019, 2020; Leveque et al., 2021; Knowles et al., 2023;





Nand and Qi, 2023). It is still unclear who will suffer the most from water deficits, which regions are most vulnerable and to what extent.

Unlike droughts, climate change studies related to flooding can rely on well-defined metrics to evaluate the areas most vulnerable to future floods and help develop adaptation strategies. This is because extreme events such as 20- or 100-year floods are frequently used in urban planning and regulatory frameworks. Infrastructure is designed to withstand specific thresholds, and maps are produced to indicate which neighbourhoods will be affected and damaged when events exceed them. In contrast, droughts emerge slowly over long periods of time, often remaining undetected in water-rich areas until they are already severe enough to significantly impact human activities. Furthermore, although Quebec has regulations regarding water withdrawals, it lacks specific low-water thresholds that would trigger emergency protocols, such as those used in the neighbouring province of Ontario (Ontario Ministry of Natural Resources et al., 2010; Berthot et al., 2020). This situation makes it difficult to predict the impacts of future severe droughts in Quebec and similarly hinders the development of effective climate change adaptation strategies.

Recently, storyline approaches have begun to be explored in climate change research and may provide a robust framework for projecting the impacts of future extreme events in the absence of clear thresholds, due to their event-based nature (Shepherd et al., 2018; Shepherd, 2019; Sillmann et al., 2021; Matte et al., 2022; Baulenas et al., 2023; Caviedes-Voullième and Shepherd, 2023). Indeed, storylines differ from traditional climate change projections, which are probabilistic, by employing climate science along with other techniques to project specific past events under future climatic conditions in a physically consistent manner. Consequently, the storyline approach focuses on the plausibility of the future scenario rather than its likelihood, and avoids the need to pinpoint specific damage-inducing thresholds (e.g., a 20-year flood) by constructing the future scenario based on an event that already caused damage in the past. Storylines have been applied in both climatological and hydrological research to effectively illustrate future extreme events such as droughts, floods and storms (Schaller et al., 2020; van der Wiel et al., 2021; Chan et al., 2022; Gessner et al., 2022).

Storylines can be constructed using various methodologies. One approach involves using a climate or weather forecast model to recreate the event and explore alternative realities, for example by modifying the initial conditions or by using a pseudo-global-warming approach (e.g., Hazeleger et al., 2015; Gessner et al., 2022; Matte et al., 2022). Another methodology involves searching for analogues of the event within a simulated dataset that includes both historical and future climate conditions (e.g., van der Wiel et al., 2021). This approach involves defining the event of interest using indicators that can accurately capture its intensity and main characteristics while also being flexible and reliable enough to identify comparable events in the simulated data. For example, in their study related to a severe drought in Europe, van der Wiel et al. (2021) examined three different types of metrics associated with water deficits: the average deficit during the most severe drought months (M1), the regression slope of the deficit in the months preceding the peak deficit (M2) and a temporal regression of the deficit time series (M3).

This study aims to employ a storyline approach similar to that described in van der Wiel et al. (2021) to project a recent drought event into future climate conditions corresponding to an increase of 2°C and 3°C in global surface temperatures compared to preindustrial levels, using analogues taken from a large ensemble of regional climate simulations. This study goes one step further than van der Wiel et al. (2021) by also making use of a hydrological model to project streamflow data during





the drought event, thus also contrasting present and future hydrological conditions. Finally, by selecting a severe drought that recently affected southern Quebec as the event of interest for this study, we are able to work alongside water management experts and other stakeholders to build a detailed understanding of the socioeconomic and environmental impacts of that event and describe the possible outcomes of climate change on future severe water shortages. This integrated approach will provide

valuable information for the development of effective adaptation strategies. Section 2 describes the data used for the analysis and the experimental framework, while Sect. 3 presents the drought event used as a reference, as well as its present and future analogues derived from a large ensemble of simulations. Finally, Sect. 4 examines the cascade of socioeconomic and environmental consequences linked to the potential outcomes of future severe water scarcity events. Concluding remarks are provided in Sect. 5.

## 2 Methodology

### 2.1 Stakeholder consultation

Documenting the impacts of previous drought events was essential to better understand the potential outcomes of future events described by the storyline approach. However, there is a known underreporting of water scarcity events in Quebec, in part due to the common belief in an abundant water supply (Nand and Qi, 2023). This is exacerbated when it comes to ecosystems, because

prevalent monitoring systems primarily focus on human water usage and rarely align with the data gathering requirements necessary to evaluate the exposure, sensitivity and adaptive capacity of wildlife and vegetation (Kovach et al., 2019; Berthot et al., 2020). Consequently, impacts on ecosystems are largely unreported unless they affect human activities.

Hence, with the objective to enhance our understanding of the impacts of previous droughts across the province, a questionnaire was circulated amongst the forty Watershed organizations (*Organisme de bassin versant*[1]) throughout Quebec. It asked

the respondent to identify up to ten recent hydrological droughts, classify their intensity and probable causes and then list the documented or perceived socioeconomic and environmental consequences. Through this process, a total of 87 water scarcity incidents were reported with varying levels of detail regarding their impacts on human and environmental usage. To complement this knowledge base, interviews were conducted with select stakeholders to deepen our understanding of past events in specific complex sectors such as the Quebec Metropolitan area or Lake Saint-Jean.

Although this consultation is not the main focus of this study (see Audet et al. (2024) – in French only – for a more detailed explanation of the methodology and breakdown of the results), certain findings will be discussed in Sect. 4. These findings provide valuable insights that enhance the understanding of the effects of previous drought events, thus enriching the discussion on the possible impacts of future droughts. By combining the qualitative data from the questionnaire with the quantitative insights drawn from the storylines, it becomes possible to obtain a more comprehensive understanding of the

potential impacts of future severe droughts.

---

[1]Independent bodies with the mandate to assess the current state of the water resource in their region and determine the priority actions that ought to be taken to improve the efficiency of water management





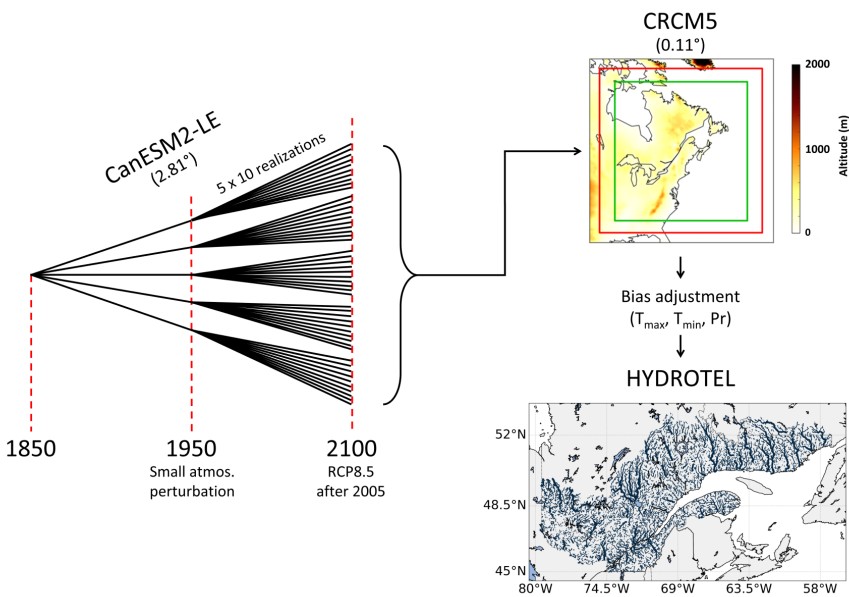

**Figure 1.** Experimental framework for the production of the ClimEx Large Ensemble (atmospheric and hydrological simulations). Fifty realizations from the CanESM2 Large Ensemble are used to drive the CRCM5 regional climate model from 1950 to 2100 using the RCP8.5 emissions scenario. Results are then bias adjusted and used in the HYDROTEL hydrological model to produce streamflow data. The bottom-right figure shows the spatial domain and river network of the Hydroclimatological Atlas of Southern Quebec. Adapted from Leduc et al. (2019).

## 2.2 Observational data

Historical precipitation and temperature data were obtained from the ERA5-Land reanalysis (Muñoz Sabater et al., 2021), which is a reconstruction of surface conditions using a 9 km spatial grid-mesh derived from the ERA5 reanalysis. Historical streamflow records were taken from the Hydroclimatological Atlas of Southern Quebec, which includes a retrospective analysis

of streamflow data for approximately 10,000 gauged and ungauged river segments from 1970 to the present (Lachance-Cloutier et al., 2017; MELCCFP, 2022). This represents all watersheds in excess of 50 km² in the southern part of the province, but leaves out river segments significantly affected by dams and other anthropogenic activities (Fig. 1, bottom right). The retrospective analysis is performed by combining streamflow measurements from various locations in the province and the output of the HYDROTEL hydrological model driven by gridded observation data, using an optimal interpolation method

(Lachance-Cloutier et al., 2017).

## 2.3 Experimental framework

The storyline approach described in van der Wiel et al. (2021) requires a significant number of climate simulations, as an adequately sized sample of simulated counterpart events is required to identify appropriate analogues. These data were obtained



from the ClimEx Large Ensemble (LE) project (www.climex-project.org; Leduc et al. (2019)), which consists of 50 perturbed
simulations produced using the fifth-generation Canadian Regional Climate Model (CRCM5; Martynov et al. (2013); Šeparović
et al. (2013)). The CRCM5 was developed by the ESCER Centre (*Centre pour l'Étude et la Simulation du Climat à l'Échelle
Régionale*) at the Université du Québec à Montréal, in collaboration with Environment and Climate Change Canada.

Temperature, specific humidity, surface pressure and winds produced by the large initial-condition ensemble (50 realizations)
of the second generation of the Canadian Earth System Model (CanESM2-LE; Fyfe et al. (2017)) were used to drive the
boundaries of the CRCM5 every 6 hours for the period 1950-2100 (Fig. 1). It used observed radiative forcings until 2005 and
the RCP8.5 emissions scenario afterwards (Riahi et al., 2011). For this experiment, the CRCM5 grid-mesh was set at 0.11°
horizontal resolution and winds were spectrally nudged from 500-hPa to the model top. For a more detailed description of the
models and setup of the ClimEx experiment see Leduc et al. (2019).

The output of these climate simulations was then bias corrected before being used as input by the HYDROTEL hydrological
model. Bias adjustment was applied to daily surface temperature and precipitation data using a Detrended Quantile Mapping
(DQM; Gennaretti et al. (2015)) technique, using 50 bins to build a transfer function linking the model to the NRCANmet
dataset. The latter is a spatial kriging of station data on a 10 km grid covering all of Canada, produced by Natural Resources
Canada (Hutchinson et al., 2009). For precipitation, the quantile mapping method was modified to combine DQM with a
Generalized Pareto distribution in the upper tail of the distribution (Roy et al., 2023). This modification was added in order to
improve the correction of the most extreme precipitation events compared to using a standard DQM technique, but should not
play a role in the results presented here, as we focus on dry conditions.

HYDROTEL is a semi-distributed physically based hydrological model used operationally for streamflow forecasting by the
government of Quebec (Fortin et al., 2001). It is partly modular in nature, with some controls on the physical representation of
channel routing, the vertical water budget and, importantly for this work, potential evapotranspiration (PET). The HYDROTEL
configuration used for this study (MG24HQ) uses the definition of PET provided by McGuinness and Bordne (1972) (see Sect.
2.4.1 for more details). The simulations were performed from 1955 onwards and generated streamflow data throughout southern
Quebec (Malenfant et al., 2024). The configuration and calibration of HYDROTEL for these simulations are identical to that
used for the historical reconstruction, guaranteeing that the simulated streamflow are consistent with the reference product.

Two future periods were selected to investigate the effects of human-induced climate change on extreme low-water episodes,
specifically targeting global average temperature rises of 2°C and 3°C above preindustrial levels. The specific periods corre-
sponding to the warming levels within each ClimEx realization were determined by using a 30-year centered moving window
to compute global 2 m temperature increases compared to the 1850-1900 period within the driving model CanESM2-LE,
locating the first instance when the 30-year window crosses these warming thresholds.





## 2.4 Drought indicators

### 165    2.4.1    Meteorological drought indicator

Various metrics have been developed to represent meteorological droughts (Palmer, 1965; McKee et al., 1993; Vicente-Serrano et al., 2010; Staudinger et al., 2014). Of these, the Standardized Precipitation Evapotranspiration Index (SPEI; Vicente-Serrano et al. (2010)) is a standardized index that measures the deviation of the water budget (difference between precipitation and evapotranspiration; $P - PET$) aggregated over a certain number of months (e.g., SPEI-$3_{AUG}$ aggregates the water budget from
June to August). It has been used in various studies both internationally and in Canada and Quebec to describe the severity and duration of droughts (Tam et al., 2019, 2023; Solaraju-Murali et al., 2019, 2021). It also notably serves as the main drought indicator on the ClimateData.ca portal, the Canadian web portal that provides national climate change information (Cannon, 2024; Lavoie et al., 2024). Consequently, the SPEI was selected as the drought indicator to represent surface conditions for this analysis. The SPEI is computed by following these steps:

1. Compute the monthly water budget ($P - PET$).

   2. Aggregate the water budget over a given number of months. In this study, aggregations of 3 and 6 months were used.

   3. For a given month, aggregation length and calibration period (e.g., water budgets aggregated from June to August over 1992-2021), fit a statistical model to the empirical distribution of water budgets and get their probability of exceedance. Based on suggestions from multiple studies, a three-parameter log-logistic distribution was employed for this
step (Stagge et al., 2015; Tam et al., 2023).

   4. Transform the probabilities of exceedance back to a normal distribution. Values below -1, -1.5 and -2 are typically used as thresholds that represent mild, severe and extreme drought conditions, respectively (McKee et al., 1993; Stagge et al., 2015).

Because the storyline approach used here requires identifying suitable analogues within a large ensemble of simulated
climate data, it was crucial to ensure that the final normal distributions would be comparable, despite potential biases in ClimEx or despite a warmer and drier future climate. Therefore, the calibration of the three-parameter log-logistic distribution and subsequent standardization was independently performed for each realization and each time period (both historical and future) to maintain consistency in the range of SPEI values.

Access to potential evapotranspiration (PET) data is required for calculating the water budget needed for the SPEI. For this
analysis, PET was calculated using the definition provided in McGuinness and Bordne (1972), which takes into account both temperature and latitude. It is defined as:

$$PET[mm\,day^{-1}] = a\frac{S_0}{\lambda}T_a + b\frac{S_0}{\lambda} \tag{1}$$

where the coefficients $a$ and $b$ represent empirical values, $S_0$ [MJ m-2 day-1] denotes the extraterrestrial solar radiation based on a solar constant of 1367 W/m$^2$, $\lambda$ stands for the latent heat of vaporization [MJ kg-1] and $T_a$ indicates the air temperature



[°C]. The values for $a = 0.0052$ and $b = 0.0875$ were adopted, as specified by Tanguy et al. (2018). This definition was chosen in order to have a consistent definition of PET with HYDROTEL, the hydrological model used in this study.

Finally, despite the standardization process, it was deemed essential to validate that the anomalies in the water budget were similar between the bias-adjusted ClimEx data and ERA5-Land, since the observed and simulated SPEIs are calculated as deviations from their respective climatology. Our findings indicate that, with the exception of the winter season where both

datasets are coherent, the water budget in the bias-adjusted ClimEx data is consistently below that of ERA5-Land. However, importantly, anomalies are similar across both datasets (Fig. S1).

### 2.4.2 Hydrological drought indicators

Various hydrological indicators were used to characterize the intensity and duration of the drought (Table 1). These indicators were specifically selected as they encapsulate crucial aspects of river conditions, thereby facilitating a better comprehension of

how extreme the historical event was across different facets of the hydrological cycle (e.g., the timing of the low-water season or the minimum summer flow). In Sect. 4, these indicators will be used in conjunction with findings from the stakeholder survey to investigate the potential links between the event's socioeconomic and environmental impacts and deviations in these indicators. Additionally, applying the same indicators and thresholds to the future events will allow for a consistent approach to assess and compare the potential impacts of future droughts, thus helping gain an understanding of how much more severe

their consequences could be.

The low-water season (LWS) described in Table 1 was defined as the period of the year when streamflow remains below $Q_{thresh}$, defined as:

$$Q_{thresh} = \overline{Q} - 0.85\left(\overline{Q} - 7Q2\right) \tag{2}$$

where $\overline{Q}$ is the annual average flow and $7Q2$ is the minimum 7-day average flow with a return period of 2 years. The LWS

begins when streamflow is below $Q_{thresh}$ for a continuous period of 7 days (after May 1st) and ends when it exceeds the threshold for 7 continuous days, such that:

$$LWS_{start} = \min\{doy \mid Q_i < Q_{thresh} \ \wedge \ \forall i \in \{doy, \dots, doy + 6\} \ \wedge \ doy > 121\} \tag{3}$$

$$LWS_{end} = \min\{doy \mid Q_i > Q_{thresh} \ \wedge \ \forall i \in \{doy, \dots, doy + 6\} \ \wedge \ doy > LWS_{start}\} \tag{4}$$

where $doy$ is the day of the year.

It should also be noted that when an indicator in Table 1 is defined using a threshold, such as $Q_{thresh}$ or $7Q2$, the threshold is defined using data from the historical period only.

Finally, the bias adjustment of ClimEx data does not guarantee that simulated streamflow data will not be biased compared to observations, since nonlinear effects might come into play in both the hydrological model and the calculation of the hydrological indicators. For the construction of the future indicators, it is therefore preferable to apply a second adjustment step to





**Table 1.** Indicators used to describe the low-water episode and its consequences

| Indicator | Definition | Use |
|---|---|---|
| $7Q_{min}$ | Minimum 7-day average flow (from May to November). | Indicates the highest stress exerted on the stream for a given year (Ismail et al., 2021). |
| $n_{days<7Q2}$ | The total number of days (from May to November) below a threshold characterized by the minimum 7-day average flow with a return period of 2 years (7Q2). | The 7Q2 is used as an 'environmental flow' in Quebec and plays a crucial role in various regulatory functions to maintain the health of river ecosystems, including setting limits for water withdrawals (Berthot et al., 2020, 2021). |
| $n_{days<7Q10}$ | The total number of days (from May to November) below a threshold characterized by the minimum 7-day average flow with a return period of 10 years (7Q10). | The 7Q10 was originally designed to protect water quality under the USA Federal Clean Water Act and represents the lowest flow necessary to preserve the dilution capacity of rivers. This indicator serves here as a metric for extreme low flow with negative consequences on water quality and ecosystems (Berthot et al., 2020, 2021). |
| $LWS_{start|end|duration}$ | Start date, end date and duration of the low-water season as defined by Eq. 2 to 4. | The start and end dates of the low water season influence various activities, including sowing schedules and fish reproduction (Morales-Marín et al., 2019; Nand and Qi, 2023). In addition, the duration of this season can indicate the severity and duration of drought conditions. |
| $\overline{Q}_{MAM|JJA|SON}$ | Average seasonal flow. | The seasonal flow anomaly provides a good estimate of the general intensity of water scarcity and can serve as a proxy for impacts on run-of-river dams. |
| $14Q_{max}$ | Maximum 14-day averaged flow. | Approximation of the freshet volume, which plays a significant role in groundwater recharge and fish reproduction (Morales-Marín et al., 2019). |

the indicators themselves (Grenier et al., 2019). For that reason, future hydrological indicators for the historical event under +2 and +3°C scenarios were constructed by calculating the average difference between the 10 best analogues in the future and historical time periods, and then applying this difference onto the observed indicator (Fig. S2). The modification was carried out by applying the relative change as a multiplier for flow-related metrics, whereas an absolute change was applied to metrics measured in days. This methodology also follows the one employed for the creation of future hydrological indicators in the

Hydroclimatological Atlas of Southern Quebec (Malenfant et al., 2024).



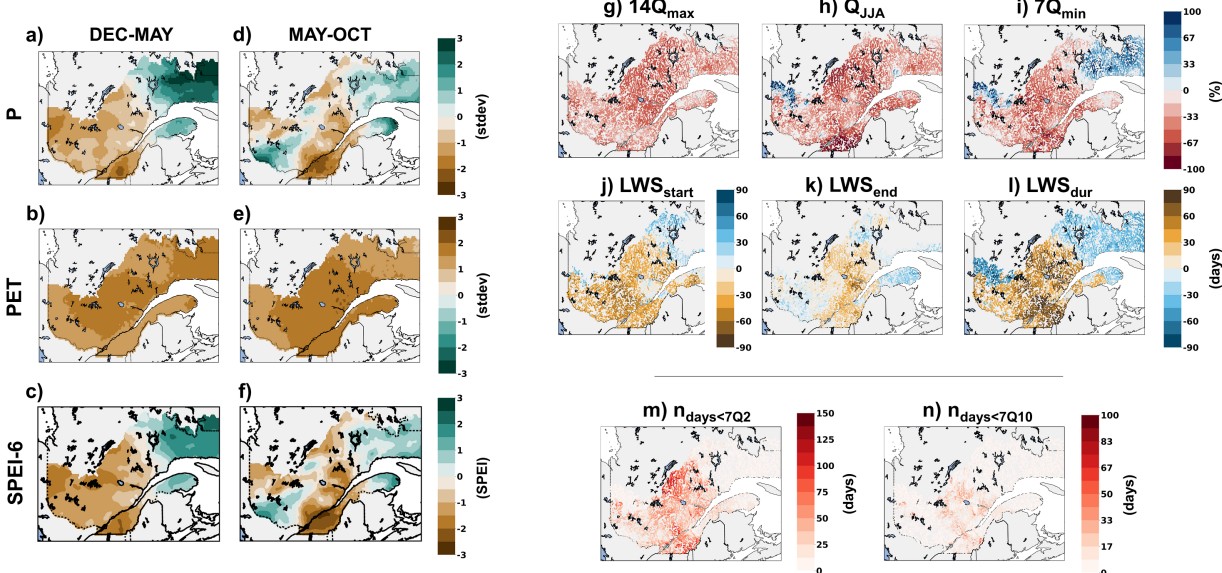

**Figure 2.** Standard deviation anomalies of precipitation (P) and potential evapotranspiration (PET) in 2021 relative to the 1992-2021 average, along with SPEI-6 values, for a-c) December to May and d-f) May to October. g-l) Anomalies in 2021 for the $14Q_{max}$, $\overline{Q}_{JJA}$, $7Q_{min}$, $LWS_{start}$, $LWS_{end}$ and $LWS_{dur}$ hydrological indicators compared to the 1992-2021 average. m-n) Absolute values reached in 2021 for $n_{days<7Q2}$ and $n_{days<7Q10}$.

## 3   Results

Years with documented severe drought conditions were identified using the stakeholder survey mentioned in Sect. 2.1, as well as an examination of ERA5-Land data. Although multiple years such as 2002, 2010, 2012 and 2018 were also highlighted by stakeholders as notably dry, the 2021 low-water event emerged as both the longest and most severe drought in recent years in

the southern parts of Quebec. As detailed in the next section, an analysis of SPEI values for that region and year showed that most of spring, summer and fall remain under severe drought conditions (SPEI ≤ -1.5) or worse. Consequently, the year 2021 was chosen as the focus of this study.

### 3.1   Analysis of the 2021 drought in Southern Quebec

Anomalies of precipitation and evapotranspiration for the year 2021, along with associated SPEI values, are shown in Fig. 2a-c

(December-May) and d-f (May-October), whereas Fig. 2g-j depict the differences between 2021 and the 1992-2021 average for key hydrological indicators taken from Table 1. Note that the climatological mean of $n_{days<7Q2}$ and $n_{days<7Q10}$ would not be meaningful, since the threshold is by definition exceeded every 2 and 10 years only. Therefore, these two indicators are presented in absolute terms rather than as anomalies.




Severe drought conditions (SPEI < -1.5) were already detected in May after an unusually warm and dry winter and spring
(Fig. 2c), leading to a reduction in freshet volumes of up to 67% below average in most of the province (Fig. 2g) and causing
the onset of summer-like conditions in rivers 45 to 60 days earlier than usual (Fig. 2j).

Drought conditions did not improve during summer and fall (Fig. 2f), with exceptionally low precipitation especially in the
St. Lawrence River valley, which, combined to high temperatures and evapotranspiration, led to extreme drought conditions
(SPEI < -2) in that region. As pointed out by Kinnard et al. (2022), the lack of summer rainfall and high evapotranspiration
led to extremely low streamflow throughout the province, but especially south of the St. Lawrence River (Fig. 2h). In that area,
which is the most densely populated of Quebec and contains most of the agricultural land, anomalies in summer flow were
as severe as -85% or more compared to the 1992-2021 average, while the rest of the province still faced streamflow at about
50% of the normal rate. The $7Q_{min}$, indicative of the maximum stress on a stream within a given year, also reached its lowest
point of the 1992-2021 period in many river sections of that region during 2021, with anomalies also ranging between -50%
and -85% in the most affected areas (Fig. 2i). Furthermore, those affected rivers experienced at least 30 days with flow below
the environmental threshold (7Q2), with some segments having over 90 days below this limit (Fig. 2m), and 14 days or more
below the 7Q10 (Fig. 2n).

Anomalies for the end of the low-water season (Fig. 2k) match the spatial pattern shown in the May-October precipitation
deficit and the $SPEI-6_{OCT}$ very well (Fig. 2e-f). Delays of up to a month can be seen in the driest areas, while substantial
rainfall in the eastern and western parts of the province, particularly in September (not depicted), alleviated drought conditions
there and helped end the LWS at a date close to the 1992-2021 average. Combined with the early onset of the LWS, the low-
water season was extended by more than two months beyond the usual duration in the areas most affected by the 2021 drought
(Fig. 2h).

The 2021 drought thus resulted in unprecedented hydrological stress in southern Quebec and around the Lake Saint-Jean
area, with significant reductions in streamflow and an extended low-water season. Despite some late seasonal precipitation
that alleviated drought conditions in some parts of the province, the south shore of the St. Lawrence River remained critically
affected, showcasing the vulnerability of this densely populated and agriculturally important area to extreme drought events.

## 3.2 Present and future analogues in the ClimEx Large Ensemble

It is crucial to identify indicators that best describe the event of interest for the storyline, since this choice can significantly
influence the analogues that are detected within the simulated dataset (van der Wiel et al., 2021). Appropriate analogues were
identified using a technique similar to the M3 method described in van der Wiel et al. (2021), which used temporal correlation.
A series of SPEI-3 indices from May through November, along with the SPEI-6 for May and October, were used as targets
during the search for analogues within the ClimEx dataset, alongside a weighting mask to assign greater importance to areas
experiencing mild to extreme droughts (see below). These weights successfully highlighted the area most affected during the
2021 drought, with a good correlation with the river sections where $Q7_{min}$ exhibited the most extreme (negative) anomalies.
This approach also proved effective for other years with significant droughts identified within the 1992-2021 period (Fig. S3).

More precisely, the identification of analogues within ClimEx was carried out through the following steps (Fig. 3):





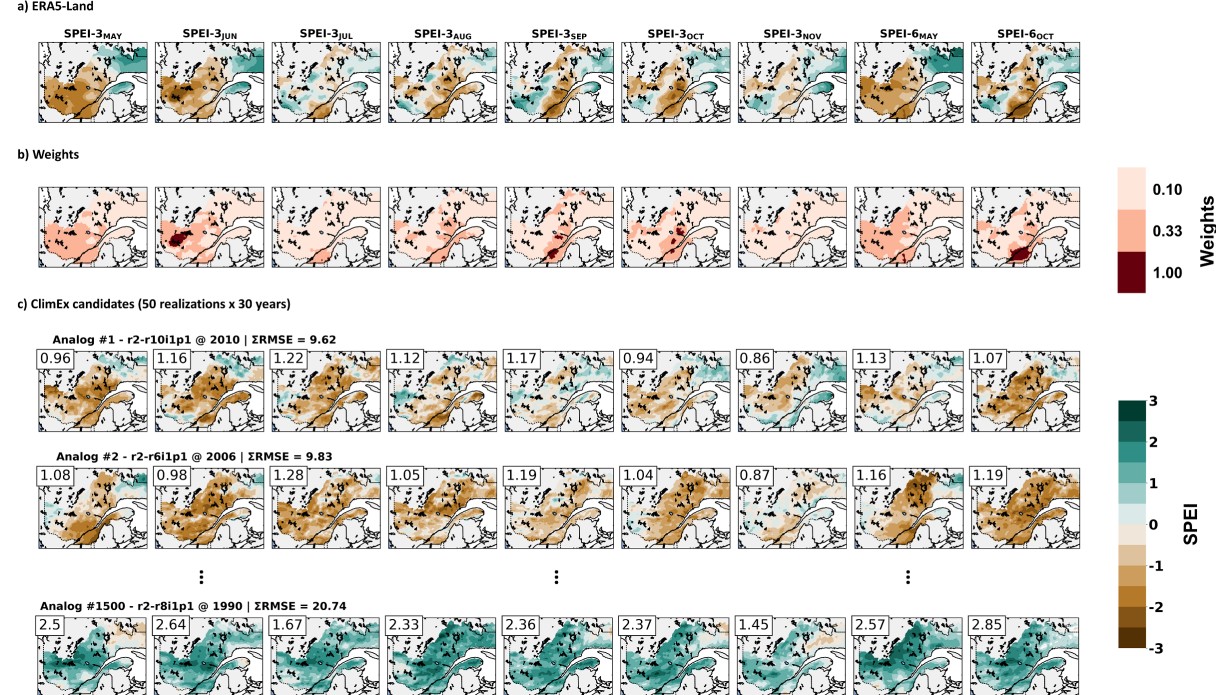

**Figure 3.** The assessment of candidate analogues is performed by comparing various SPEI indicators (columns) between a) ERA5-Land and c) 1500 candidates (50 realizations x 30 years) from the ClimEx Large Ensemble. For each candidate (rows in c), a weighted RMSE is calculated for each SPEI (top-left corner of each image) and then aggregated. Weights (b) are allocated based on SPEI values in ERA5-Land.

1. Calculate the SPEI-3 for all months from May to November, and the SPEI-6 for May and October, using the year 2021 for ERA5-Land and using 1500 candidates over a 30-year span for ClimEx (50 realizations x 30 years).

2. Apply a weighting mask based on the categories specified by McKee et al. (1993) for each SPEI value derived from ERA5-Land to emphasize lower values, namely:

   (a) 1 if SPEI $\leq$ -2 (extreme drought)

   (b) 0.33 if SPEI $\leq$ -1 (mild drought)

   (c) 0.1 for all other values

3. Calculate a weighted root mean square error (RMSE) for each grid cell within the domain, comparing the SPEI values from ERA5-Land against each of the 1500 ClimEx candidates.

4. Sum the RMSEs for each ClimEx candidate.

The 1500 possible candidates within ClimEx for each period (historical, +2°C and +3°C) were then sorted according to their total RMSEs, cumulated over the spatial domain of HYDROTEL and the 9 SPEI indicators analyzed (Fig. 3). The 10



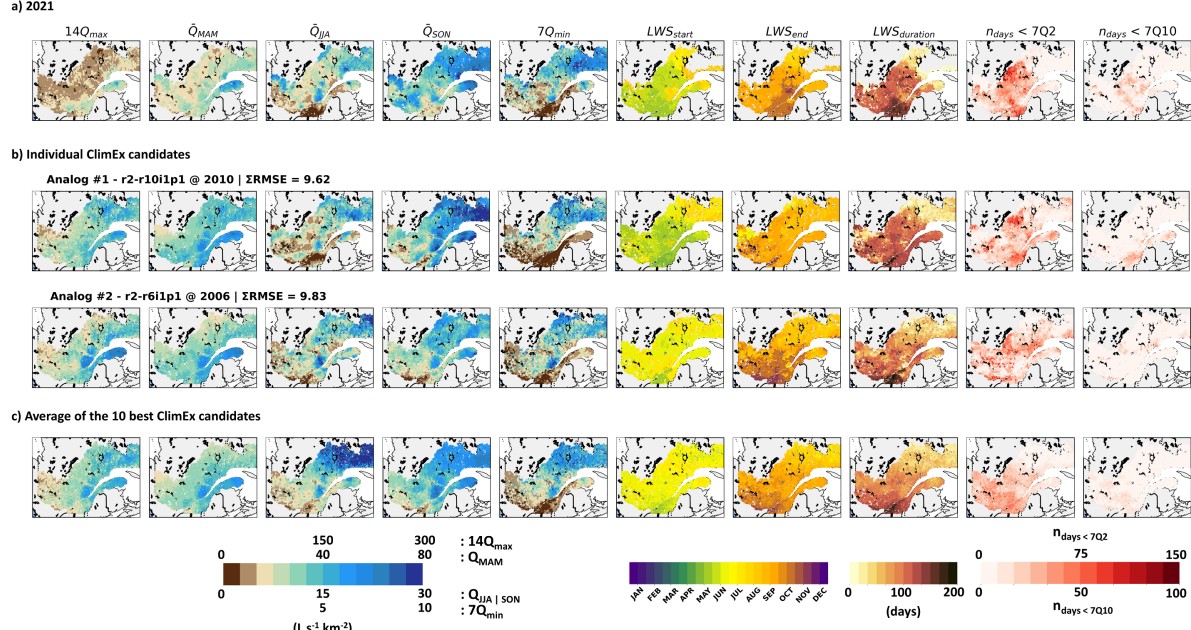

**Figure 4.** Comparison of hydrological indicators between (a) the 2021 low-water episode, (b) the first two best ClimEx candidates within the historical period and c) the average of the 10 best ClimEx candidates.

candidates with the lowest total RMSE were selected as the 10 best analogues to 2021. This step was accomplished separately for the historical and the future periods.

Figure 4 presents a comparison of the hydrological indicators for the 2021 event, for the top two analogues of 2021 in the historical period and for the average of the 10 best analogues. The average of the 10 best analogues (Fig. 4c) replicates the spatial patterns and intensity of the 2021 drought particularly well for key indicators like $7Q_{min}$ or $LWS_{duration}$, although it overestimates the freshet volume and spring flow. It also adequately reproduces other indicators such as $\overline{Q}_{JJA|SON}$, $LWS_{start|end}$ and $n_{days<7Q2|7Q10}$, though individual members sometimes provide a more accurate match in specific instances. For example, while the best match (*r2-r10i1p1* in Fig. 4b) excels in replicating $LWS_{start}$, it ends too early for $LWS_{end}$. The second best match (*r2-r8i1p1* in Fig. 4b) exhibits the opposite issue. Therefore, on the whole, the average of the 10 best analogues performs adequately well and offers a solid representation of the 2021 drought for the chosen hydrological indicators (Fig. 4).

## 3.3 Future hydrological droughts

Future hydrological indicators were produced by averaging the 10 best analogues in ClimEx, as detailed in Sect. 2.4.2 and illustrated in Fig. S2. Figures 5 and 6 resemble Fig. 2, but they illustrate the change in the 2021 drought under the +2°C and +3°C warming scenarios relative to the historical event. More detailed versions for all hydrological indicators are shown in the Supplementary Materials (Fig. S4 to S13).



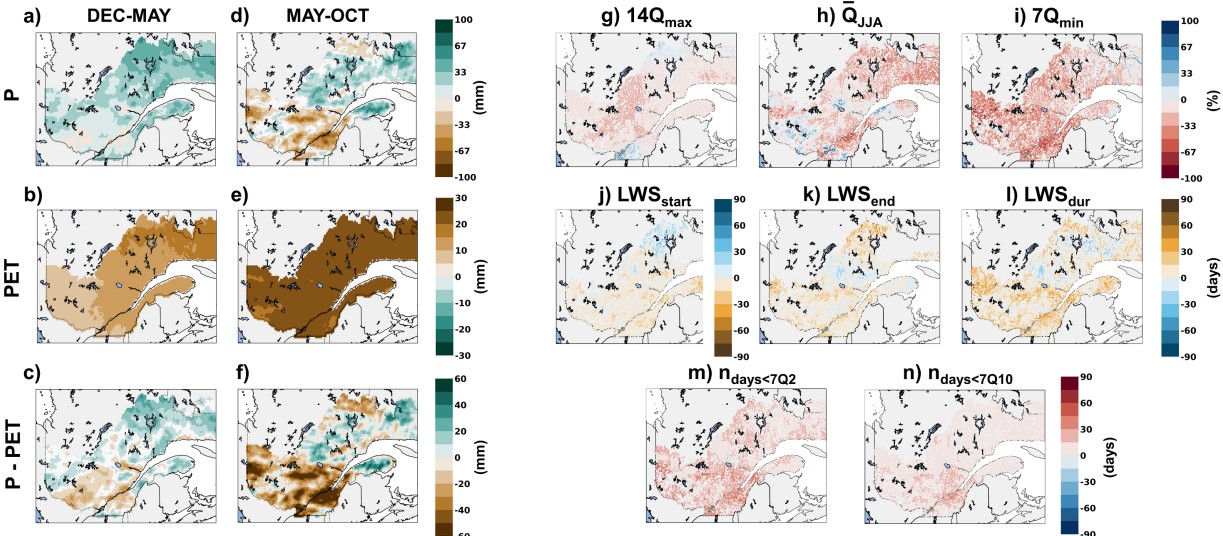

**Figure 5.** Precipitation (P) and potential evapotranspiration (PET) anomalies under the +2°C scenario with respect to the 2021 conditions, along with the anomalies in the water budget ($P - PET$) for a-c) December to May and d-f) May to October. g-n) Anomalies for the hydrological indicators in the +2°C scenario with respect to the 2021 conditions.

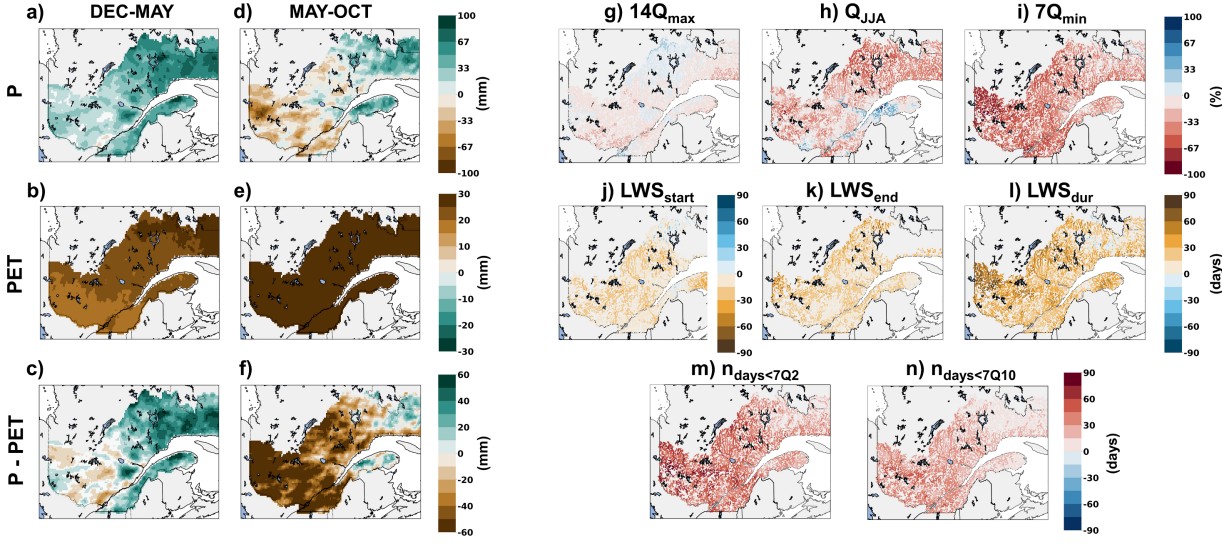

**Figure 6.** Same as Fig. 5, but for the +3°C scenario.

The 2021 event under +2°C warming conditions shows a rise in winter and spring precipitation compared to the historical event (Fig. 5a), particularly in the northern regions, in accordance with IPCC projections (IPCC, 2023). This increase in precipitation is offset by an increase in PET throughout the province (Fig. 5b), resulting in a slight reduction in the water



budget in southern areas and a minor increase in northern regions (Fig. 5c). As a result, spring freshet volumes remain mostly within ±15% of 2021 levels (Fig. 5g), while the start date of the low-water season also remains similar to 2021 (Fig. 5j).

Conditions during summer and fall are generally drier than 2021, especially in the southern and western parts of the province (Fig. 5d), although September still features significant rainfall in both the eastern and western parts of the province (not shown). Due to higher temperatures, the future scenario also features a substantial increase in evapotranspiration that affects the whole province (Fig. 5e). This leads to a net deficit in water balance that affects all areas that had suffered from the 2021 drought, but now also reaches the western parts of the province that had been less impacted during the historical event (Fig. 5d). Rainfall

in September had contributed to ending the low-water season between late September and mid-October during the 2021 event. As this rainfall episode still occurs in the future scenario, the LWS ends at a similar time of the year (Fig. 5k). Combined with a similar start in the LWS with respect to 2021, this means that the duration of the LWS is either the same as in 2021, or only slightly longer (Fig. 5l).

 As a result of the drier conditions, the average streamflow in summer for many river segments shows an additional reduction

of up to 33% compared to 2021, although a small improvement can be noted south and west of the province where precipitation is higher than in 2021 (Fig. 5h). The $7Q_{min}$ was already significantly below average in 2021, and the analogues indicate a further decline of this metric across the entire province, with an additional reduction of 15-50% compared to 2021 (Fig. 5i). As a result, under the +2°C scenario, the total number of days below the environmental flow threshold ($n_{days<7Q2}$) increases by 15 to 30 days for numerous river segments (Fig. 5m).

The 2021 event under +3°C warming conditions is generally in agreement with the +2°C scenario, but shows even stronger reductions in streamflow and water availability across the province due to rising temperatures and evapotranspiration. The only exception is in winter and spring, where a further increase in precipitation leads to a rise in the net water balance for most of the province compared to 2021 (Fig. 6c). However, this does not result in a higher $14Q_{max}$, which remains similar to the historical event (Fig. 6g) potentially due to the increase in precipitation being balanced by more winter thaw and higher

winter streamflow. This could also explain why the low-water season for the analogue under a +3°C scenario begins 2-4 weeks earlier than in 2021 (mid-May compared to early June) for numerous river sections north of the St. Lawrence River (Fig. 6j), especially since precipitation is still well below the historical average despite the future increase.

 Precipitation changes in summer and fall under the +3°C warming conditions closely resemble those in the +2°C scenario, but PET increases are significantly greater and lead to a net water balance loss of more than 60 mm over most of the southern

and western parts of the province (Fig. 6f). This causes a further reduction in the average seasonal streamflow across the province (Fig. 6h) and also negatively affects the $7Q_{min}$ metric, which displays a reduction of 33-67% compared to the already low levels of 2021 (Fig. 6i). The rise in $n_{days<7Q2}$ is thus very pronounced under the +3°C scenario, with some segments experiencing an increase of up to 60 days (Fig. 6m). Given the already high number of days recorded in 2021, this implies that many rivers could experience up to 100 days below the environmental flow threshold, with some potentially reaching 150

days. Under that scenario, it is particularly concerning that the more severe low-flow metric (7Q10) is exceeded by most rivers across the province for a duration of 25 to 60 days (Fig. 6n). As discussed in Sect. 4, this would suggest unprecedented impacts on ecosystems, water quality and human uses.



Finally, as a consequence of both the decrease in precipitation and the significant rise in evapotranspiration, the LWS concludes up to a month later compared to 2021 for many river sections under +3°C conditions, bringing low-water conditions into

early November in the southern parts of the province (Fig. 6k). Under that scenario, there is a total increase of 15 to 45 days of the LWS throughout most of the province compared to 2021, as a result from both an earlier onset and a later conclusion of the LWS (Fig. 6l).

## 3.4 Limitations

We recognize that there are a certain number of limitations to the current analysis. The use of a single hydrological model,

namely HYDROTEL, is one of them. It is well known that using multiple hydrological models provides a better representation of the uncertainty (Castaneda-Gonzalez et al., 2022). Furthermore, HYDROTEL is primarily geared toward surface water dynamics and employs simplified representations of soil and groundwater processes and interactions. Although the methodology used in this study to generate future indicators – which involves applying the climate change signal onto the reconstructed streamflow data – partially mitigates this limitation, HYDROTEL cannot reproduce low-flow conditions as well as fully inte-

grated surface-groundwater models.

Another constraint on simulating future low flows is HYDROTEL's current dependence on a relatively simplistic definition of PET, which relies only on near-surface temperature (McGuinness and Bordne, 1972). Although other definitions of PET are available in HYDROTEL, they all rely on that single variable. Internal testing has shown the climate change signal to be relatively insensitive to the selection of PET among the currently available definitions (not shown), but this conclusion might

differ if more physically-based formulas that included variables like solar radiation or humidity were available for comparison.

However, it is essential to note that the primary objective of this study was not to achieve exact modelling of future low flows, but to produce plausible, physically coherent low-flow indicators reminiscent of a recent event to stimulate conversations with stakeholders about the effects of water scarcity in future climates in southern Quebec. The constructed scenarios offer valuable insights into potential future conditions and emphasize the importance of preparedness for water shortages. Additionally, as

outlined in the subsequent section, these scenarios lay the groundwork for dialogues on the environmental and socioeconomic repercussions of such an event and underscore the necessity of implementing robust water management strategies to ensure the sustainability and resilience of water resources in Quebec.

Although this study mainly examines water availability, it must be noted that future projections occur within a framework of anticipated increased water demand, which, in Quebec, is primarily driven by population growth (Charron et al., 2019, 2020;

Da Silva et al., 2020; Achkar, 2023). The province is well-known for its high per capita daily water consumption and the residential sector alone currently represents 41% of the province's water usage (Charron et al., 2019). Based on data from Institut de la Statistique du Québec (2022), the population is projected to increase by 6% by 2030 relative to 2021, reaching 13% by 2050 and 17% by 2066. This growth could further strain the already scarce resources during severe droughts like that of the analogues presented in this study. However, several strategies are already being implemented to reduce water consumption

and bring it closer to the Canadian average, which might help mitigate part of the increase in demand resulting from population growth in the coming decades (Government of Quebec, 2019).





## 4 Discussion

Water availability is critical and, similarly to other vital resources, its scarcity progressively affects the health of communities and populations through a cascading series of impacts (Parisi et al., 2018; Bond et al., 2019). Climate change, which modifies
the climate system and causes a reduction in water availability during summer due to increasing temperatures, could trigger a series of effects, starting with the deterioration of habitats for plants and animals reliant on water, resulting in a loss of biodiversity and the ecological services offered by wetlands and river systems (Mehdi et al., 2015; Furber et al., 2016; Wheaton and Kulshreshtha, 2017; Osman, 2018; Chilima et al., 2021). These alterations subsequently impact human activities because of the diminished quality and availability of water, eventually influencing the well-being of the population. Furthermore, water
scarcity can lead to competition over resource usage and conflicts between different purposes, whether for human needs or ecological functions.

The discussion presented in this section is inspired by the results of the stakeholder survey (Sect. 2.1). Notably, because the survey was event-centric in nature, in the sense that it asked to report the consequences of the droughts for each event individually, it supported the storyline approach and helped investigate the cascade of consequences that occurred in 2021 and
during other droughts. It also facilitated the extrapolation of those consequences into future climate conditions by employing descriptors for the future events that were directly connected to 2021 (e.g. "an analogous event under a +3°C temperature increase would last 30 days longer and have its $7Q_{min}$ reduced by half"). Although we recognize that these findings are anecdotal in nature, these results constitute a valuable first step that offers perspectives on the impacts of historical droughts and help initiate a dialogue on water shortages in Quebec.

### 4.1 Environmental consequences


Despite the survey, impacts on ecosystems were difficult to identify unless they had affected human activities. Even when relevant data were available, it was mostly punctual and anecdotal in nature. This is in line with other studies that have noted a lack of consistent data collection before, during and after drought periods that could help build causal relationships between low-water episodes and their impacts on ecosystems (Kovach et al., 2019). Even environmental flow metrics have been deemed
insufficient to truly represent the needs of ecosystems, with ongoing efforts in Quebec to replace the 7Q2 indicator with another indicator better suited to the geomorphological conditions of rivers and more robust in the context of a changing climate (Berthot et al., 2020, 2021).

Therefore, the effects of the 2021 drought on ecosystems, when analyzed through the lens of the storyline approach, must still be viewed from a qualitative standpoint. The persistently low flow throughout the year, ranging from reduced seasonal
streamflow to exceptionally low $7Q_{min}$, all indicate deteriorated water quality conditions (e.g., reduced pollutant dilution, higher water temperatures, decreased oxygen availability, eutrophication and lower pH levels in some rivers) (Zohary and Ostrovsky, 2011; Dormoy-Boulanger, 2021). During the survey, multiple stakeholders, especially in the area south of the St. Lawrence River, either directly acknowledged a decline in water quality in 2021 or indirectly highlighted this issue by noting higher treatment costs for municipalities or mass fish mortality events. Furthermore, while only a few Watershed organiza-





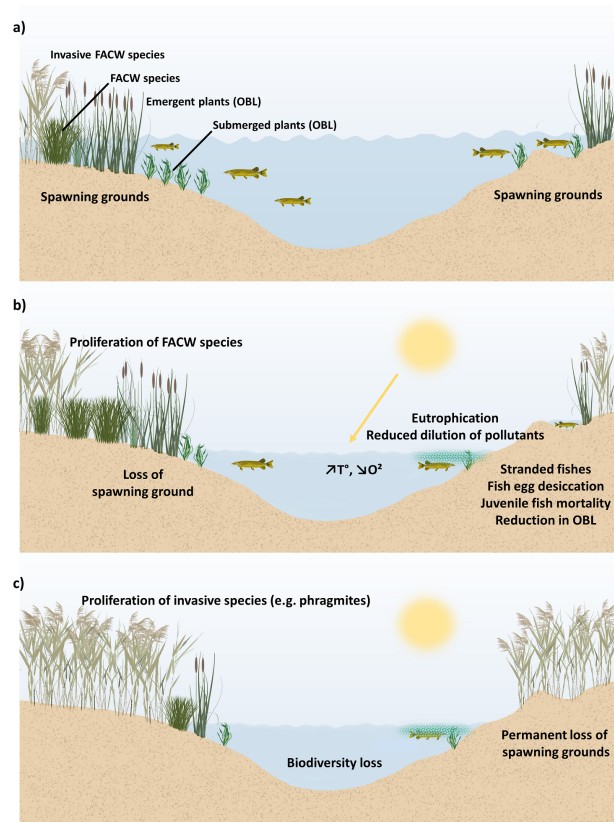

**Figure 7.** Depiction of river ecosystems. a) Under normal spring conditions, b) under diminished flow conditions and c) additional effects from severe or extended low-flow conditions. Facultative Wetland species (FACW) are flora that typically occur in wetlands but can be found elsewhere on occasion, while Obligate Wetland species (OBL) exist almost exclusively in wetlands. Adapted from Audet et al. (2024).

tions reported it, it was noted that the decrease in freshet volume and the early arrival of low-water levels adversely affected submerged plant species to the benefit of invasive alien plant species such as *Phragmites australis*. This can be problematic, since native aquatic plants often provide crucial shelter for numerous species of fish and serve as spawning grounds, so their disappearance could lead to reduced fish populations and biodiversity (Boyer et al., 2009; Plourde-Lavoie et al., 2017; Hudon et al., 2016; Mejia et al., 2023). Furthermore, a brief and weak freshet could result in fish being trapped in the floodplain or

having no access to spawning areas, in elevated juvenile fish mortality rates and in the desiccation of fish eggs (Plourde-Lavoie et al., 2017). Figure 7 provides a diagrammatic illustration of the impacts of low-water episodes on plant and fish species, comparing normal spring conditions to diminished flow conditions and severe or extended low-flow conditions.

     As mentioned earlier, except for the freshet volume, all low-flow indicators become notably worse under +2°C and +3°C conditions, suggesting the possibility of severe ecological degradation if extreme drought conditions similar to 2021 were to

occur again in a warmer climate. Although the 7Q2 has been criticized as an insufficient benchmark for assessing ecosystem





needs (e.g., Berthot et al., 2020), the significant rise in the number of days falling below this threshold and the 7Q10, particularly under a +3°C rise in global temperatures, is worrying. This situation could further strain already vulnerable ecosystems in a future climate, with potential long-term impacts on biodiversity and ecosystem services. The combined effects of higher temperatures and extended droughts could cause irreversible alterations to aquatic habitats, further endangering the survival of

native species and the overall health of the ecosystem (Fig. 7c).

## 4.2 Socioeconomic consequences

Recent studies, as well as results from the stakeholder survey, indicate that some municipalities and some economic sectors in southern Quebec have become more vulnerable to water scarcity in recent years (Charron et al., 2019, 2020; Leveque et al., 2021; Knowles et al., 2023; Nand and Qi, 2023). Some cities now face recurrent challenges in providing drinking water to

their population in summer, even under normal conditions, while others have had to restrict new developments due to potential water scarcity (Audet et al., 2024). In Quebec, almost 40% of the population is supplied with surface water from rivers or lakes, excluding the St. Lawrence River which accounts for another 30%, while groundwater provides the remaining 30% (MELCCFP, 2018). Although municipalities sourcing their water directly from the St. Lawrence River have not experienced severe water shortages, the St. Lawrence River valley itself, which is the province's most densely populated area with extensive

agricultural lands, is already under significant stress for both surface and groundwater resources (e.g., Carrier et al., 2013; Larocque et al., 2015). As previously highlighted by the SPEI-6$_{OCT}$ and hydrological metrics like the 7Q$_{min}$ and LWS$_{duration}$, that region was also the most impacted by the 2021 drought (Fig. 2). Consequently, in 2021, numerous municipalities relying on surface water reported difficulties due to decreased water availability and quality. This situation prompted them to issue boil water advisories, bear increased water treatment costs, limit non-essential water usage and, in one instance, set up a pumping

system to draw additional water from the Lake of Two Mountains (where the Ottawa River delta meets the St. Lawrence River, near Montreal). Municipalities dependent on groundwater also had to impose similar usage restrictions to conserve their water supply, while some individuals and farms relying on private wells had to either deepen their existing wells or dig new ones (Audet et al., 2024).

In the agricultural sector, over 55% of water is used for livestock and fish farming, while the remainder is distributed among

the cleaning of buildings, the washing of fruits and vegetables and irrigation. Notably, 95% of Quebec's agricultural land is allocated to field crops and hay, with the majority being rainfed, whereas irrigation is primarily limited to valuable crops like cranberries and various other fruits and vegetables (Charron et al., 2019, 2020; Nand and Qi, 2023). Consequently, the 2021 drought resulted in reduced crop yields throughout the agricultural industry and, although the economic impact of the 2021 drought on the agricultural sector has not yet been quantified, previous droughts resulted in financial losses amounting to tens

of millions of dollars (Nand and Qi, 2023). Furthermore, according to the survey, some farmers faced additional significant expenses, as they had to employ water tanker trucks to replenish basins and provide water to their livestock (Nand and Qi, 2023; Audet et al., 2024).

The recreotourism industry also experienced several impacts. In numerous regions, streamflow was exceptionally low for the entire summer, leading to decreased water levels in rivers and lakes. This had a detrimental effect on marinas, sometimes





severely limiting boating activities. In smaller rivers, low water levels also hindered the use of canoes, kayaks and paddleboards. Swimming was also restricted due to poor water quality (Audet et al., 2024). Finally, while the conditions in 2021 likely caused losses in industries such as pulp and paper production or hydroelectric generation, no public information on that topic could be found.

As mentioned previously, all low-flow indicators suggest worsening conditions for the future 2021 analogues, indicating
additional strain and adverse effects on municipal water supply systems, agriculture, recreational tourism and various other sectors. The 2021 drought highlighted the vulnerability of Quebec's water resources and the potential for widespread consequences. The magnitude of the future drought as described by the storyline approach suggests that the impacts could be even more significant, necessitating urgent and comprehensive water management strategies that do not yet exist in the regulatory framework of the province to address these challenges effectively. In fact, future analogues show even lower seasonal stream-
flow and $7Q_{min}$ than in 2021, while the LWS in the St. Lawrence Valley begins earlier and ends later, indicating a longer and harsher summer season that could further deplete available water resources.

## 5   Conclusions

This study used a storyline approach to discuss the plausible impacts of severe drought conditions in Quebec based on a recent event. By analyzing the evolution of SPEIs from spring to fall, analogous events were identified within a large ensemble of
climate simulations, both in present and future climates, offering a comprehensive understanding of future drought conditions akin to those of 2021 in two different warming level scenarios.

These storylines can provide a framework for stakeholders to gain a clearer understanding of future droughts, by linking them to an event that they are already familiar with. Indeed, by building a storyline based on the 2021 drought, we were able to effectively explore the potential consequences of similar future events under increases in global temperature of +2°C and
+3°C. However, there is a need for improved systematic data collection with regards to the impacts of meteorological and hydrological droughts on human activities and even more so on ecosystems in Quebec and Canada. Research is also required on the effects of climate change on groundwater and its recharge, particularly during extreme droughts, as there is currently a scarcity of literature in this area despite the central role of aquifers in the water supply of Quebec. Enhancing the groundwater monitoring network, both spatially and temporally, is essential to address these research deficiencies and generate the data
required to shape effective water management policies.

Lacking a comprehensive understanding of these factors, the implementation of adaptation strategies could become significantly more challenging or may even result in maladaptive outcomes (e.g. increasing the reliance on aquifers without an assessment of how vulnerable they are to extreme droughts). Yet, without measures to improve resilience against water shortages in Quebec, a region typically abundant in water but lacking adequate regulatory systems to manage water usage disputes,
it is anticipated that various socioeconomic and environmental consequences will occur. Potential impacts may include a reduction in biological diversity and deterioration of ecosystem services, diminished agricultural yields or greater dependence



on irrigation, increased expenses for water purification, a halt of residential expansion in areas with scarce water resources and a rise in disputes over water-related issues.

However, it is also important to recognize the adaptive capacity of ecosystems and the potential for recovery. With proper management and conservation efforts, the resilience of aquatic environments can be enhanced. Initiatives such as restoring river connectivity and improving water quality can mitigate some of the negative impacts observed during low-water periods. Moreover, increasing awareness and involving local communities in water conservation practices offers a hopeful perspective towards sustaining biodiversity and ecosystem services even under challenging conditions.

Future studies should focus on integrating more localized data and predictive models to refine the accuracy of drought projections and their impacts, as well as gaining a better understanding of the climatic causes behind hydrological and groundwater droughts in the region. Collaborative efforts between climatologists, hydrologists, biologists and policy makers are essential to develop more robust adaptation strategies that can be implemented effectively. Furthermore, exploring the environmental and socioeconomic implications of droughts through interdisciplinary research (Matte et al., 2024) will provide deeper insights into the cascading effects of water shortages and inform better regulatory and community response strategies. Such proactive approaches are crucial to mitigating the adverse effects of future droughts and improving the resilience of communities in Quebec and similar regions around the world.

*Code availability.* The code used in this research can be found on GitHub at the following link: https://github.com/Ouranosinc/CASCADES/

*Author contributions.* G.R.G. was the lead author of the manuscript, aided by L.P.C.; É.C. initially examined the storyline approach with help from G.R.G., L.P.C., and D.M., which was later advanced by G.R.G. who also performed the climatological and hydrological analyzes. K.A. managed the stakeholder survey and contributed to the socioeconomic evaluation, which was led by L.d.A.; D.T. and R.P. led the environmental impact assessment. All authors provided crucial feedback and helped refine the research, analysis and manuscript.

*Competing interests.* The authors declare that they have no competing interests.

*Acknowledgements.* This project was made possible thanks to funding from the *Gouvernement du Québec*. The authors also want to thank Isabelle Charron, Diego Crespel, Bertrand Montel, Matthieu Paccard and Jérémie Roques for their help and support throughout the CASCADES project. We acknowledge the scientific monitoring committee of the CASCADES project (Anne Blondlot, Richard Turcotte, Nathalie Bleau, Charlotte Legault-Bélanger, Marie Larocque, Annick Van Campenhout, Sébastien Ouellet-Proulx and Frédéric Lecomte) for their valuable insights.

ERA5-Land was downloaded from the Copernicus Climate Data Store (CDS). The results contain modified Copernicus Climate Change Service information.



The production of ClimEx was funded within the ClimEx project by the Bavarian State Ministry for the Environment and Consumer Protection. The CRCM5 was developed by the ESCER centre of Université du Québec à Montréal (UQÀM; www.escer.uqam.ca) in collaboration with Environment and Climate Change Canada. We acknowledge Environment and Climate Change Canada's Canadian Centre for Climate Modelling and Analysis for executing and making available the CanESM2 Large Ensemble simulations used in this study. Computations with the CRCM5 for the ClimEx project were made on the SuperMUC supercomputer at Leibniz Supercomputing Centre (LRZ) of

the Bavarian Academy of Sciences and Humanities. The operation of this supercomputer is funded via the Gauss Centre for Supercomputing (GCS) by the German Federal Ministry of Education and Research and the Bavarian State Ministry of Education, Science and the Arts.

Streamflow data was obtained from the Hydroclimatological Altas of Southern Quebec, which is produced by the *ministère de l'Environnement, de la Lutte contre les changements climatiques, de la Faune et des Parcs* of Quebec in collaboration with Ouranos. The Hydroclimatic Atlas is produced with the support of numerous collaborators, and its development is financially supported by the *Fonds vert* as part of the im-

plementation of the Quebec government's *Plan d'action 2013-2020 sur les changements climatiques* and the *Plan pour une économie verte 2030*.

Proofreading for the manuscript was assisted by AI technologies through Writefull.



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
