# Peer review of "Storyline Analytical Framework for Understanding Future Severe Low-Water Episodes and Their Consequences"

_EGUsphere, 2024_

## Referee Comment (RC1)

*Review comments on the article submitted to HESS (egusphere-2024-2595) entitled:*

Storyline Analytical Framework for Understanding Future Severe Low-Water Episodes and Their Consequences

by

Gabriel Rondeau-Genesse, Louis-Philippe Caron, Kristelle Audet, Laurent Da Silva, Daniel Tarte, Rachel Parent, Élise Comeau, and Dominic Matte

This is an interesting and well written article that presents a storyline approach for analyzing future drought conditions in the southern part of the province of Quebec, Canada. The basis for the analysis is the drought event of 2021, for which, in the implemented storyline methodology, similar events (so called 'analogues') are identified in the future, under two Climate Change (CC) scenarios of 2 °C and 3°C degrees average warming. Future climate is simulated using an ensemble of fifty perturbed simulations in which data from a Canadian global circulation model CanESM2 are used to drive a regional climate model CRCM5 for the period 1950-2100 using the RCP8.5 (high emissions) scenario adopted by IPCC. The analogues are identified based on an existing methodology proposed in (van der Wiel et al. 2021), using Standardized Precipitation-Evapotranspiration Index (SPEI) as main indicator of similarity between the 2021 event and the future analogues (historical analogues for the period 1992-2021 have also been identified and subsequently used). The methodology is then extended by using hydrological drought indicators (streamflow-related), which are proposed and calculated for a large number of Southern Quebec watersheds using a hydrological model, driven by bias-corrected outputs of the regional climate model. Main results are presented as maps with spatial distribution of these indicators for current conditions (2021 event) and for conditions coming from the average of the 10 best analogues (most similar to 2021 events), which are then used to assess potential drought impacts on ecosystems and socio-economic activities in the region. The article is suitable for publication in HESS, after addressing the following comments:

1. As the calculation of the hydrological drought indices is central for providing the results of this research, it is necessary to provide some more information about the HYDROTEL hydrological model, and how it was used / set up for this research. Readers unfamiliar with this model need some basic information about the concepts used in it, which then needs to be extended with the following: How many watersheds were actually simulated? (There is a mentioning of 10,000 watersheds and a threshold of 'larger than 50km$^2$', but it is not clear how many were finally simulated; all maps with indicator results show streams, but it is not clear how are they related to the simulated watersheds). Which exact outputs of the regional climate model were used? Were these only precipitation and temperature (for which the implemented bias corrections are discussed)? Were the watershed models set up with or without sub-basins? How were inputs and parameters specified (gridded or per sub-basin)? Which were dominant runoff components simulated (snowmelt? direct runoff? subsurface runoff (base flow, which would be important for drought conditions?)? In the 'Limitations' section the authors mention some deficiencies of

the hydrological model used, and that the value of their article is primarily in the methodological framework that has been introduced, and in "producing plausible, physically coherent low-flow indicators" However, the above suggested information, in my view, is still needed for better understanding of the hydrological modelling component of this work.

2. There is certain disconnection between the analytical work carried out to provide the results in terms of hydrological drought indices and the discussion part presented in section 4, especially sections 4.1 and 4.2. These sections mostly present information about drought impacts on ecosystems and socio-economic activities during the 2021 event. There are only some suggestions about how some of the calculated hydrological drought indices could be used to better assess drought impacts, but most of the content in these sections is a description of the impacts from 2021 drought.  My suggestion would be to somewhat re-arrange the content in this article and to bring in the description of the 2021 drought in terms of impacts on ecosystems and socioeconomics much earlier (under 'Introduction', or under 'Methodology' as a separate section, or somehow combined with the section on 'Stakeholder consultation'). The argument can then be introduced that lack of adequate hydrological drought indices to assess future droughts under CC is an issue that this article addresses (There is even mentioning that the province of Quebec indeed does not have such indicators in line 60). The whole content of calculating these indices would then follow, and the Discussion section can be somewhat extended about the value / potential use of the proposed indicators. The need to move towards the next step, which is developing impact-related drought indicators could then be emphasized in that discussion. This is just a suggestion, and the authors may decide on a different approach to better connect the discussion part with the previous part of the article.

Specific comments:

1. Line 5, in the abstract: The statement "This approach allowed for enhanced collaboration with water management experts and other stakeholders to project the possible impacts of climate change on serious water deficits in Quebec." should be modified or removed. There is no evidence presented in the article that this actually occurred. (This comment is somewhat related to the general comment 2, mentioned above).

2. Please insert somewhere in Section 1 (Introduction) or Section 2 (Methodology) a figure with an overview map of the actual study area, together with geographical features to which you are later on referring (e.g. St. Lawrence river / valley, Ottawa river, Lake of Two Mountains, etc.). This will be very useful for readers unfamiliar with the geography of Quebec.

3. Please make it clear in Section 1 (Introduction) that the focus of this research is hydrological drought (and the indices). When one reads this section, it is not always clear whether meteorological or hydrological drought will be the target of the analysis (later on it becomes clear).

4. Line 60: Please add "and ecosystems" at the end of the sentence "…significantly impact human activities", to be consistent with the content presented later.

5. Line 85: Please do not use 'water deficits', as it adds to the ambiguity regarding what is the focus of the cited study and of this study. The cited study is clearly about precipitation deficits

(meteorological drought) and this study goes further in calculating hydrological drought indices based on streamflow data.

6. Line 95: Please us 'Sect.' or 'Section' consistently.

7. Line 115: The statement "By combining the qualitative data from the questionnaire with the quantitative insights drawn from the storylines, it becomes possible to obtain a more comprehensive understanding of the potential impacts of future severe droughts" should be modified or removed. There is no evidence in the content of the article that such combination was performed. If it is a suggestion that this should be done in the future – the statement should be re-formulated. (This comment is related to the general comment 2, and specific comment 1, mentioned above).

8. Line 125: the sentence starting with "The retrospective analysis…" is not clear. Could you please expand it or re-formulate it? What is meant by 'combining streamflow measurements from various locations'? What was combined and how? Which 'gridded observation data were used for driving HYDROTEL' (again, see general comment 1)?

9. Line 150: The statement "… and, importantly for this work, potential evapotranspiration (PET)" is a bit surprising. Practically all hydrological models use PET as an input, so how can this be a particular feature of HYDROTEL that is important?

10. Line 175: This approach is not entirely clear to me. The SPEI values will be comparable in terms of values, but if they come from two different distributions (from two different climates), how can then they be comparable? Can you please elaborate a bit?

11. Line 225: So, there were 10 best analogues for the historical period *and* 10 best analogs for the future, and the difference between the averages of the two was calculated, right? Please clarify.

12. Lines 270 and 280: Would the results be sensitive to these weights values? Has this been tested? If not, perhaps it should be recommended?

13. Section 3.4 (Limitations). Perhaps it could be recommended to carry out similar analysis for individual basins, and not only for such a large area? This would bring the use of the indicators closer to actual water resources planning and management, in my view…

14. Figure 7. I don't see why this figure is necessary. Many drought impacts have been discussed, and not supported by similar generic information (and figures). Why is this generic view on drought impacts on river systems chosen to be presented? It is a bit of a distraction, in my view. I would suggest to present relevant figures for this particular research (perhaps Figure S2 from the supplementary material?)

Thank you very much for an interesting article.

---

## Referee Comment (RC2)

**Review of "Storyline Analytical Framework for Understanding Future Severe Low-Water Episodes and Their Consequences" by Rondeau-Genesse et al.**

**Summary**

This study presents an interesting analysis of streamflow droughts in Quebec using a storyline approach. The authors sampled for low-flow and severe drought events similar to the 2021 drought based on a range of metrics within hydrological simulations (HYDRTEL) driven by the CRCM5-LE regional climate model large ensemble. The authors find that drought analogues in a 2°C and 3°C world are drier with lower streamflow compared to 2021. This approach is a powerful way to communicate future changes in drought severity by anchoring analysis on a recent observed event and this study is a valuable contribution to an emerging body of literature on the creation of storylines for hydrological extremes. This article is well-suited for HESS and should be considered for publication after addressing the following comments. I hope my comments will help the authors improve their paper.

**Comments**

1. Bias-adjustment of climate model: The authors mention that the detrended quantile mapping bias adjustment technique was applied for precipitation. It isn't clear whether the bias adjustment was applied to each ensemble member independently from each other? It is recommended that any parameters for bias adjustment applied to single-model-initial-condition large ensembles should be computed from the pooled ensemble rather than individual ensemble members to preserve the range of internal variability in the original ensemble. It would be good if the authors can clarify their approach.

2. Hydrological model: more details of the hydrological model would be appreciated. For example, was the model calibrated to observed streamflow in any way and what is the spatial resolution of the hydrological model? Were there any statistical downscaling methods applied to further downscale the large ensemble data prior to hydrological modelling? Related to this, would the authors be able to provide some indication of model performance compared to observed flows at gauges (e.g. in the supplement)? There should also be some discussion of the simulated river flows driven by ClimEx over the historical period – do they exhibit similar hydrolgical behaviour to observed river flows and where does observed flows lie within the wider range of ClimEx simulated flows?

3. SPEI calculation: Have the authors looked at how their results might change if the SPEI distribution for the future periods are fitted to the parameters calculated from the historical period? This could serve as a good sensitivity test for whether the decision to calculate SPEI separately for historical and future period is a valid approach.

4. Changes in drought analogues: as the authors note, there are interesting hydrological dynamics within the future analogues (such as changes in winter thaw and changes in the timing of low flow season) which deserves a bit more discussion. Could the authors provide a time series of P, PET and simulated flows for the baseline observed 2021 and for the top 10 future drought analogues (either at particular selected catchments or averaged)? The readers can then visualise these temporal changes easier.

5. Discussion: The current Discussion needs to be reframed. The section seems out of place and reads more like a separate literature review rather than an actual discussion of the results. Perhaps the authors could reframe this to link to the results of the paper (e.g. how these storylines can be used to further drive water quality models or how flow indicators from the drought analogues could be used to infer different drought impacts). Much of the socio-economic impacts of the 2021 drought could actually go in the introduction to highlight the severity of the event and motivate why the authors decided to choose the 2021 event as a

case study. There should also be further discussion of the utility of the storyline approach and future work (e.g. are there limitations to the storyline approach? How can such an approach be used alongside other climate change projection products?)

**Minor amendments**

- IPCC (2023) should be cited according to the official suggested citation according to the IPCC:

  Lee, J.-Y., J. Marotzke, G. Bala, L. Cao, S. Corti, J.P. Dunne, F. Engelbrecht, E. Fischer, J.C. Fyfe, C. Jones, A. Maycock, J. Mutemi, O. Ndiaye, S. Panickal, and T. Zhou, 2021: Future Global Climate: Scenario-Based Projections and NearTerm Information. In Climate Change 2021: The Physical Science Basis. Contribution of Working Group I to the Sixth Assessment Report of the Intergovernmental Panel on Climate Change [Masson-Delmotte, V., P. Zhai, A. Pirani, S.L. Connors, C. Péan, S. Berger, N. Caud, Y. Chen, L. Goldfarb, M.I. Gomis, M. Huang, K. Leitzell, E. Lonnoy, J.B.R. Matthews, T.K. Maycock, T. Waterfield, O. Yelekçi, R. Yu, and B. Zhou (eds.)]. Cambridge University Press, Cambridge, United Kingdom and New York, NY, USA, pp. 553–672, doi:10.1017/9781009157896.006.

- L80 – The authors may consider expanding the literature search and also refer to more recent studies which have also used an analogue approach to sample for drought/heatwave storylines – The following three studies may be useful, the first two for meteorological droughts and the third for hydrological droughts:
  1. Faranda, D., Pascale, S., and Bulut, B.: Persistent anticyclonic conditions and climate change exacerbated the exceptional 2022 European-Mediterranean drought, Environ. Res. Lett., https://doi.org/10.1088/1748-9326/acbc37, 2023.
  2. Liao, Z., An, N., Chen, Y., and Zhai, P.: On the possibility of the 2022-like spatio-temporally compounding event across the Yangtze River Valley, Environ. Res. Lett., 19, 014063, https://doi.org/10.1088/1748-9326/ad178e, 2024.
  3. Chan, W. C. H., Arnell, N. W., Darch, G., Facer-Childs, K., Shepherd, T. G., Tanguy, M., and van der Wiel, K.: Current and future risk of unprecedented hydrological droughts in Great Britain, Journal of Hydrology, 130074, https://doi.org/10.1016/j.jhydrol.2023.130074, 2023.

- L108 and elsewhere – there are several mentions of a questionnaire that was given to stakeholders to gather information on the 2021 drought and to disseminate results. More discussion of what the outcomes from this questionnaire should be given. Could the questionnaire be published in the supplement as well for transparency?

- L155 – what is MG24HQ?

- Font sizes in all figures could be bigger. Consider showing a subset of the results and putting other sub-plots in the supplementary materials. For example, Figure 3 could be simplified by including SPEI-3 averaged over May-Oct rather than each month separately (the monthly plots could go in the supplementary materials).

- The Limitations section should go within the Discussion section rather than Results.

---

## Referee Comment (RC3)

[referee-annotated manuscript omitted]

---

## Author Comment (AC1)

We would like to thank the referees for their constructive feedback. We appreciate the time and effort that was put into the review and we hope that our responses prove satisfactory. For clarity, the reviewer's comments are presented in black, with our responses in red.

N.B. The Atlas' technical report was cited as Malenfant et al., 2024 in the first version of the manuscript, but this will be changed to DPEH, 2024.

Best regards,

Gabriel Rondeau-Genesse, on behalf of all authors.

**Referee #1**

**General comments:**

**1-1. As the calculation of the hydrological drought indices is central for providing the results of this research, it is necessary to provide some more information about the HYDROTEL hydrological model, and how it was used / set up for this research. Readers unfamiliar with this model need some basic information about the concepts used in it, which then needs to be extended with the following: How many watersheds were actually simulated? (There is a mentioning of 10,000 watersheds and a threshold of 'larger than 50km², but it is not clear how many were finally simulated; all maps with indicator results show streams, but it is not clear how are they related to the simulated watersheds). Which exact outputs of the regional climate model were used? Were these only precipitation and temperature (for which the implemented bias corrections are discussed)? Were the watershed models set up with or without sub-basins? How were inputs and parameters specified (gridded or per sub-basin)? Which were dominant runoff components simulated (snowmelt? direct runoff? subsurface runoff (base flow, which would be important for drought conditions?)? In the 'Limitations' section the authors mention some deficiencies of the hydrological model used, and that the value of their article is primarily in the methodological framework that has been introduced, and in "producing plausible, physically coherent low-flow indicators" However, the above suggested information, in my view, is still needed for better understanding of the hydrological modelling component of this work.**

We will clarify the text in that section and provide additional details on the hydrological model setup, based on the information below.

HYDROTEL (Fortin et al., 2001) is a distributed hydrological model that uses computational units called "Relatively Homogeneous Hydrological Units" (RHHUs). These units are determined based on land use, soil classification, and geographical characteristics such as slope and aspect. The average area of RHHUs in the Atlas is 11 km², with an associated hydrological network that consists of 28,035 river segments. For a given timestep, the streamflow at each river segment is the sum of incoming streamflow from upstream segments and runoff/baseflow from neighboring RHHUs. Although all 28,035 river segments are modeled, results are published for only 9,665 segments, based on the list of criteria mentioned in our manuscript. The vast majority of rejected river segments are due to the watershed size criterion.

The specific details regarding model configuration and the calibration setup for the Atlas are described in DPEH, 2024 (link, in French). A regional calibration was performed to optimize the average performance (calculated with a KGE objective function) across 70 watersheds, with validation over 151 stream gauges. The calibration includes two parameter sets—one for the north and one for the south of the St. Lawrence River—,

each covering 35 watersheds. Six calibration strategies/model configurations were used. The full list of parameters and their values is available in Annex D of the Atlas' technical report (DPEH, 2024). Due to computational constraints, the "MG24HQ" configuration, recommended by the Government of Quebec for its superior performance in summer flows, was selected for our study (see Annex E of the report for validation results).

HYDROTEL is a physically based hydrological model, so all runoff components are simulated explicitly. The specific formulae can vary between model configuration and are listed in Table 16 (Annex D) of the technical report. The subsurface model, BV3C, is relatively simple and uses a cascade of three subsurface reservoirs, which is why that aspect was mentioned in the limitations section.

It will be made clearer that the only meteorological inputs for HYDROTEL are daily minimum and maximum temperature, and precipitation. Other state variables, such as snow and evapotranspiration, are calculated by the model itself.

**1-2. There is certain disconnection between the analytical work carried out to provide the results in terms of hydrological drought indices and the discussion part presented in section 4, especially sections 4.1 and 4.2. These sections mostly present information about drought impacts on ecosystems and socio-economic activities during the 2021 event. There are only some suggestions about how some of the calculated hydrological drought indices could be used to better assess drought impacts, but most of the content in these sections is a description of the impacts from 2021 drought. My suggestion would be to somewhat re-arrange the content in this article and to bring in the description of the 2021 drought in terms of impacts on ecosystems and socioeconomics much earlier (under 'Introduction', or under 'Methodology' as a separate section, or somehow combined with the section on 'Stakeholder consultation'). The argument can then be introduced that lack of adequate hydrological drought indices to assess future droughts under CC is an issue that this article addresses (There is even mentioning that the province of Quebec indeed does not have such indicators in line 60). The whole content of calculating these indices would then follow, and the Discussion section can be somewhat extended about the value / potential use of the proposed indicators. The need to move towards the next step, which is developing impact-related drought indicators could then be emphasized in that discussion. This is just a suggestion, and the authors may decide on a different approach to better connect the discussion part with the previous part of the article.**

We agree that there is a disconnect between the results and the discussion. This comment and the suggestion are coherent with Comment #2-5. We will adjust the sections of our paper based on these recommendations.

**Specific comments:**

**1-3. Line 5, in the abstract: The statement "This approach allowed for enhanced collaboration with water management experts and other stakeholders to project the possible impacts of climate change on serious water deficits in Quebec." should be modified or removed. There is no evidence presented in the article that this actually occurred. (This comment is somewhat related to the general comment 2, mentioned above).**

We agree that this might have been overstated. We will modify that line accordingly.

**1-4. Please insert somewhere in Section 1 (Introduction) or Section 2 (Methodology) a figure with an overview map of the actual study area, together with geographical features to which you are later on**

referring (e.g. St. Lawrence river / valley, Ottawa river, Lake of Two Mountains, etc.). This will be very useful for readers unfamiliar with the geography of Quebec.

We will either modify Figure 1 to provide a bigger map of the study area, complemented with the relevant geographical features, or add a new figure. In the latter case, this would be in the Supplements.

**1-5. Please make it clear in Section 1 (Introduction) that the focus of this research is hydrological drought (and the indices). When one reads this section, it is not always clear whether meteorological or hydrological drought will be the target of the analysis (later on it becomes clear).**

We will add information to that effect in the last paragraph of the introduction.

**1-6. Line 60: Please add "and ecosystems" at the end of the sentence "…significantly impact human activities", to be consistent with the content presented later.**

This will be changed accordingly.

**1-7. Line 85: Please do not use 'water deficits', as it adds to the ambiguity regarding what is the focus of the cited study and of this study. The cited study is clearly about precipitation deficits (meteorological drought) and this study goes further in calculating hydrological drought indices based on streamflow data.**

This will be changed accordingly.

**1-8. Line 95: Please us 'Sect.' or 'Section' consistently.**

We will rework that sentence to circumvent the use of both forms, but this followed the journal guidelines: "The abbreviation "Sect." should be used when it appears in running text and should be followed by a number unless it comes at the beginning of a sentence."

**1-9. Line 115: The statement "By combining the qualitative data from the questionnaire with the quantitative insights drawn from the storylines, it becomes possible to obtain a more comprehensive understanding of the potential impacts of future severe droughts" should be modified or removed. There is no evidence in the content of the article that such combination was performed. If it is a suggestion that this should be done in the future – the statement should be re-formulated. (This comment is related to the general comment 2, and specific comment 1, mentioned above).**

That sentence will be removed or modified, depending on its state after the restructuration of the paper proposed in the Comment #1-2.

**1-10. Line 125: the sentence starting with "The retrospective analysis…" is not clear. Could you please expand it or re-formulate it? What is meant by 'combining streamflow measurements from various locations'? What was combined and how? Which 'gridded observation data were used for driving HYDROTEL' (again, see general comment 1)?**

The term "retrospective analysis" might not have been clear or appropriate. That sentence will be reworked alongside with the additions proposed in the Comment #1-1. Here are more details:

The historical streamflow data in the Hydroclimatic Atlas is reconstructed from 1970 to the present using six simulations (the six calibration strategies mentioned in our response to #1-1) of the HYDROTEL hydrological model driven by gridded observation data. These modeled streamflow are then corrected using an optimal interpolation method, which incorporates gauged streamflow data to adjust the modeled results, accounting

for both model and observational uncertainties (Lachance-Cloutier et al., 2017). This process is similar to data assimilation techniques used in meteorological reanalyses.

**1-11. Line 150: The statement "... and, importantly for this work, potential evapotranspiration (PET)" is a bit surprising. Practically all hydrological models use PET as an input, so how can this be a particular feature of HYDROTEL that is important?**

This will be rephrased. The main point is that HYDROTEL provides several options for computing Potential Evapotranspiration (PET), and that since PET is central to our methodology, the choice of computation method could potentially impact our results significantly (though, as stated at L.355, it does not). The intention was not to suggest that this is a particularly unique feature of HYDROTEL.

**1-12. Line 175: This approach is not entirely clear to me. The SPEI values will be comparable in terms of values, but if they come from two different distributions (from two different climates), how can then they be comparable? Can you please elaborate a bit?**

We will revise that section to clarify our intentions.

By performing two calibrations—one for the past and one for the future—we can identify events that have a similar chance of occurrence and a similar progression throughout the year in both time periods. This allows us to link the historical extreme event (2021) to similarly rare and extreme future events. In flood terms, this would be akin to comparing the current 100-year flood to the future 100-year flood. Using two distributions makes this comparison more intuitive and meaningful.

**1-13. Line 225: So, there were 10 best analogues for the historical period *and* 10 best analogs for the future, and the difference between the averages of the two was calculated, right? Please clarify.**

We will clarify this sentence, but this is correct.

**1-14. Lines 270 and 280: Would the results be sensitive to these weights values? Has this been tested? If not, perhaps it should be recommended?**

We will add a clarification on this topic. Multiple tests were indeed conducted on the weighting scheme. Our results show that while the selected events themselves are sensitive to the weights and that the outcomes are thus not robust with a limited selection (e.g., 3 best analogues), the results are stable and robust when 10 or more analogues are used.

**1-15. Section 3.4 (Limitations). Perhaps it could be recommended to carry out similar analysis for individual basins, and not only for such a large area? This would bring the use of the indicators closer to actual water resources planning and management, in my view…**

We can include recommendations on this in the limitations, as well as in the discussion if it is relevant after the restructuring proposed in the Comment #1-2. While we agree that this provides a broad overview, the distributed nature of HYDROTEL means that indicators are nonetheless computed at a local level, which we hope will have actionable value. A second phase of the project will begin in early 2025, focusing on further exploring the use of these storylines at the local scale, in collaboration with stakeholders in key watersheds affected by the 2021 drought.

**1-16. Figure 7. I don't see why this figure is necessary. Many drought impacts have been discussed, and not supported by similar generic information (and figures). Why is this generic view on drought impacts on river**

systems chosen to be presented? It is a bit of a distraction, in my view. I would suggest to present relevant figures for this particular research (perhaps Figure S2 from the supplementary material?)

We deemed that figure to be useful as a generic portrayal of the impacts of droughts on river ecosystems, given that a typical reader of this paper is likely to be less knowledgeable in that field. However, with the restructuration of that section proposed in the Comment #1-2, it is likely that this figure will have to be either moved to the Supplements or removed, since the associated text will be moved earlier.

We will consider the addition of Figure S2 to the main body.

**Referee #2**

**General comments:**

**2-1. Bias-adjustment of climate model: The authors mention that the detrended quantile mapping bias adjustment technique was applied for precipitation. It isn't clear whether the bias adjustment was applied to each ensemble member independently from each other? It is recommended that any parameters for bias adjustment applied to single-model-initial-condition large ensembles should be computed from the pooled ensemble rather than individual ensemble members to preserve the range of internal variability in the original ensemble. It would be good if the authors can clarify their approach.**

A clarification will be brought to the text. The 50 members were indeed pooled together prior to bias adjustment.

**2-2. Hydrological model: more details of the hydrological model would be appreciated. For example, was the model calibrated to observed streamflow in any way and what is the spatial resolution of the hydrological model? Were there any statistical downscaling methods applied to further downscale the large ensemble data prior to hydrological modelling? Related to this, would the authors be able to provide some indication of model performance compared to observed flows at gauges (e.g. in the supplement)? There should also be some discussion of the simulated river flows driven by ClimEx over the historical period – do they exhibit similar hydrolgical behaviour to observed river flows and where does observed flows lie within the wider range of ClimEx simulated flows?**

We will provide more details on the model setup. Refer to our responses to Comments #1-1 and #1-10 for answers to some of the questions raised.

While we likely won't include specific validation results in the paper, we will provide a more direct reference to that information when reworking the section. All validation results, including the KGE of the model and the RMSE of the minimum 7-day flow (7Qmin), are available in Annex E of the Atlas' technical report (DPEH, 2024; link, in French).

No additional spatial downscaling was applied to ClimEx data, as its spatial resolution (0.11°) is already close to the regridded observations used as a reference (10 km).

Direct comparisons between observed flow and simulated river flows driven by ClimEx are challenging due to the optimal interpolation method applied to historical flow (see our response to Comment #1-10 for more details). Figure 4 shows that there is a good agreement in river flows between observations and ClimEx

analogues for 2021, but a few additional figures comparing the river flows between ClimEx and the observations over specific river segments could be added to the Supplements.

**2-3. SPEI calculation: Have the authors looked at how their results might change if the SPEI distribution for the future periods are fitted to the parameters calculated from the historical period? This could serve as a good sensitivity test for whether the decision to calculate SPEI separately for historical and future period is a valid approach.**

Tests were conducted early in the project, but the results were difficult to apply effectively to the focus of our work (i.e., identifying future extremes that are more severe but still relatable to historical events). Please also refer to our response to Comment #1-12.

**2-4. Changes in drought analogues: as the authors note, there are interesting hydrological dynamics within the future analogues (such as changes in winter thaw and changes in the timing of low flow season) which deserves a bit more discussion. Could the authors provide a time series of P, PET and simulated flows for the baseline observed 2021 and for the top 10 future drought analogues (either at particular selected catchments or averaged)? The readers can then visualise these temporal changes easier.**

We will consider adding such figure(s). Averaging over an area as large as Southern Quebec is likely meaningless for such an analysis, so we will select a few catchments that were impacted in 2021.

**2-5. Discussion: The current Discussion needs to be reframed. The section seems out of place and reads more like a separate literature review rather than an actual discussion of the results. Perhaps the authors could reframe this to link to the results of the paper (e.g. how these storylines can be used to further drive water quality models or how flow indicators from the drought analogues could be used to infer different drought impacts). Much of the socio-economic impacts of the 2021 drought could actually go in the introduction to highlight the severity of the event and motivate why the authors decided to choose the 2021 event as a case study. There should also be further discussion of the utility of the storyline approach and future work (e.g. are there limitations to the storyline approach? How can such an approach be used alongside other climate change projection products?)**

This comment and the proposed solution align with Comment #1-2. We will restructure the sections accordingly.

**Minor amendments:**

**2-6. IPCC (2023) should be cited according to the official suggested citation according to the IPCC:**

> Lee, J.-Y., J. Marotzke, G. Bala, L. Cao, S. Corti, J.P. Dunne, F. Engelbrecht, E. Fischer, J.C. Fyfe, C. Jones, A. Maycock, J. Mutemi, O. Ndiaye, S. Panickal, and T. Zhou, 2021: Future Global Climate: Scenario-Based Projections and NearTerm Information. In Climate Change 2021: The Physical Science Basis. Contribution of Working Group I to the Sixth Assessment Report of the Intergovernmental Panel on Climate Change [Masson-Delmotte, V., P. Zhai, A. Pirani, S.L. Connors, C. Péan, S. Berger, N. Caud, Y. Chen, L. Goldfarb, M.I. Gomis, M. Huang, K. Leitzell, E. Lonnoy, J.B.R. Matthews, T.K. Maycock, T. Waterfield, O. Yelekçi, R. Yu, and B. Zhou (eds.)]. Cambridge University Press, Cambridge, United Kingdom and New York, NY, USA, pp. 553–672, doi:10.1017/9781009157896.006.

Noted. This will be changed.

**2-7. L80 – The authors may consider expanding the literature search and also refer to more recent studies which have also used an analogue approach to sample for drought/heatwave storylines – The following three studies may be useful, the first two for meteorological droughts and the third for hydrological droughts:**

> Faranda, D., Pascale, S., and Bulut, B.: Persistent anticyclonic conditions and climate change exacerbated the exceptional 2022 European-Mediterranean drought, Environ. Res. Lett., https://doi.org/10.1088/1748-9326/acbc37, 2023.

> Liao, Z., An, N., Chen, Y., and Zhai, P.: On the possibility of the 2022-like spatio-temporally compounding event across the Yangtze River Valley, Environ. Res. Lett., 19, 014063, https://doi.org/10.1088/1748-9326/ad178e, 2024.

> Chan, W. C. H., Arnell, N. W., Darch, G., Facer-Childs, K., Shepherd, T. G., Tanguy, M., and van der Wiel, K.: Current and future risk of unprecedented hydrological droughts in Great Britain, Journal of Hydrology, 130074, https://doi.org/10.1016/j.jhydrol.2023.130074, 2023.

Thank you for the references. These will be consulted and added to the introduction.

**2-8. L108 and elsewhere – there are several mentions of a questionnaire that was given to stakeholders to gather information on the 2021 drought and to disseminate results. More discussion of what the outcomes from this questionnaire should be given. Could the questionnaire be published in the supplement as well for transparency?**

As the results of the questionnaire were primarily used for the discussion, how we address this comment will depend on the restructuring proposed in the Comments #1-2 and #2-5. If it is still relevant, we will provide more information on the outcomes and some of the results.

We can add the questionnaire to the Supplements.

**2-9. L155 – what is MG24HQ?**

We will rephrase it to be clearer. "MG24HQ" is the name of the specific calibration of HYDROTEL that was used for our study. Refer to our response to Comment #1-1 for details.

**2-10. Font sizes in all figures could be bigger. Consider showing a subset of the results and putting other sub-plots in the supplementary materials. For example, Figure 3 could be simplified by including SPEI-3 averaged over May-Oct rather than each month separately (the monthly plots could go in the supplementary materials).**

Font sizes will be increased. For Figure 3, we can hide some of the SPEI-3 data and present the full figure in the Supplements. However, averaging SPEIs is generally not recommended. For other figures, we will simplify them where possible, though all indicators that are mentioned in the text should be present in the paper itself.

**2-11. The Limitations section should go within the Discussion section rather than Results.**

This section was placed in the Results due to the nature of the Discussion section. However, with the proposed restructuring, this will no longer be an issue, and it will be moved as suggested.

**Referee #3**

**General comments:**

**3-1. The appropriateness of the approach is claimed, but hardly explained and/or compared with other approaches. Why can the extensive modelling approach add to the baseline method? Why would it be needed (or even allowed) to keep the storyline approach when running such models? The discussion might have revealed some of these issues, but paragraph 4 does not really engage with the climate stories/narratives as such.**

The storyline approach is introduced in lines 65-85 of the Introduction, with specific examples from the literature. Some of the vagueness (as noted in the commented PDF) arose from the lack of established terminology for the various methods used to create storylines. However, a recent paper by Riglos et al. (2024) has since further categorized different types of climate storylines and narratives (https://www.sciencedirect.com/science/article/abs/pii/S1462901124001825). "Physical Climate Storylines" (PCS) (e.g., Hazeleger et al., 2015; Gessner et al., 2022; Matte et al., 2022) recreate past events in a modeling setup, then project those events into future climate conditions by modifying them within the model. Our methodology, along with that of van der Wiel et al. (2021), falls under "Scenario-based Approaches" (SBA). Instead of recreating and modifying past events using a climate model, SBAs involve identifying similar events within existing climate simulations. To find adequate analogues to the real event, SBAs typically rely on large ensembles.

This part of the Introduction will be reworked to incorporate Riglos et al. (2024) and other papers referenced by Referee 2.

**3-2. The text actually reads much like a rather traditional text on climate scenarios. It might be that the use of the model enforces this, but I found it a little tricky to find the added value of storylines - or even the difference with other ways of sharing scenarios. I would think that the text could be much the same without ever mentioning the term "storyline analysis".**

As mentioned in the previous comment, storylines aim to extrapolate specific events, such as the 2021 drought, into future climate conditions to analyze potential differences in impacts. Another way to view them is as "temporal analogues." This approach contrasts with the more traditional method of producing climate scenarios through probabilistic ensembles of multiple climate models. Please also refer to my response to Comment #1-12.

We will revise the Discussion as recommended by all Referees to better highlight the value of the method and the potential applications of the proposed indicators.

**3-3. The method section seems to be incomplete. First, the method of storyline analytics is not discussed. How does one do it? Second, in the results section there seem to be several series of steps included that might need to be included in the method section.**

How this is addressed will depend on the restructuring of the text, but Section 3.2 could be moved to the Methodology. Additionally, as suggested by Referee 1, Figure S2 will likely be moved from the Supplementary Materials to the main body of the paper to visually support the methodology used to construct the future analogues. See Comments #1-2 and #2-5.

**3-4. The usefulness of the approach is defended by claiming that the results can be used in discussions with stakeholders, with some details on stakeholders ideas in section 4 (which for some reason is presented as discussion and not as results). Any evidence for that claim is not offered, as the stakeholder data come from a survey that was done before the storyline process. As such, we read two different parts: one on climate scenarios and one on survey results. I could also imagine that in a dialogue with stakeholders, modelling scenarios, whether they are shared in storyline format or otherwise, might change the stories themselves. This can only be checked after engaging with the stakeholders. I would be very interested in that aspect.**

This issue has also been noted in Comments #1-2 and #2-5, concerning the disconnect between our results and the discussion. By reworking the paper as suggested by both referees, we aim to clarify the added value of the storyline approach. The discussion will be revised to better align with our results and their potential applications. Please also refer to our response to Comment #1-15.

**3-5. In conclusion, I would think that this text is a little unbalanced in terms of explaining the usefulness of the method, in terms of added value of the modelling, and in terms of the rather different section 3 and 4. Section 4 does seem to miss the point on discussing results from Section 3 anyway. The storyline itself reads like a standard climate scenario explanation, which is useful, but NOT necessarily new enough to be published.**

The restructuring of the paper should improve the flow of the text and reduce the disconnect between Sections 3 and 4.

**Specific comments (from the PDF file):**

L.5

We agree that this might have been overstated. We will modify that line accordingly.

L.25

We can rephrase that sentence to clarify that the issue is not that droughts in warm regions are easy to analyze (they are not), but rather that snow-driven recharge introduces an additional layer of complexity in cold regions.

L.70

Refer to the answer to Comment #3-1.

L.125

That section will be reworked to include more information on the model (see the response to Comment #1-1) and the sentence will be clarified. "Rivers significantly influenced by dams and other anthropogenic activities" refers to rivers where streamflow is actively managed on a day-by-day basis. Passive weirs and dams managed monthly are still included in the Atlas. While this does exclude several rivers, these are typically larger rivers with higher mean streamflow, making problematic low water levels less likely to occur (though not impossible). In any case, these rivers had to be excluded, as HYDROTEL is currently unable to account for them.

L.130

This will be rephrased. The term "significant" was used to indicate that a large number of simulations are needed (the higher, the better), rather than in its statistical sense.

L.185

Clarifications will be added to the text. Please refer to our response to Comments #1-12.

L.470

We agree that this might have been overstated. We will modify that line accordingly.

---

## Author Response (AR1)

We would like to thank the referees for their constructive feedback. We appreciate the time and effort that was put into the review and we hope that our responses prove satisfactory. For clarity, the reviewer's comments are presented in black, with our responses in red.

Best regards,

Gabriel Rondeau-Genesse, on behalf of all authors.

**Referee #1**

**General comments:**

**1-1. As the calculation of the hydrological drought indices is central for providing the results of this research, it is necessary to provide some more information about the HYDROTEL hydrological model, and how it was used / set up for this research. Readers unfamiliar with this model need some basic information about the concepts used in it, which then needs to be extended with the following: How many watersheds were actually simulated? (There is a mentioning of 10,000 watersheds and a threshold of 'larger than 50km², but it is not clear how many were finally simulated; all maps with indicator results show streams, but it is not clear how are they related to the simulated watersheds). Which exact outputs of the regional climate model were used? Were these only precipitation and temperature (for which the implemented bias corrections are discussed)? Were the watershed models set up with or without sub-basins? How were inputs and parameters specified (gridded or per sub-basin)? Which were dominant runoff components simulated (snowmelt? direct runoff? subsurface runoff (base flow, which would be important for drought conditions?)? In the 'Limitations' section the authors mention some deficiencies of the hydrological model used, and that the value of their article is primarily in the methodological framework that has been introduced, and in "producing plausible, physically coherent low-flow indicators" However, the above suggested information, in my view, is still needed for better understanding of the hydrological modelling component of this work.**

Section 2.2 (Observational data) has been substantially reworked to include more information on HYDROTEL and the configuration used in this study. All the points raised in this comment are now addressed.

*"Historical streamflow data were obtained from the Hydroclimatological Atlas of Southern Quebec, which covers the period from 1970 to the present and spans the entire region of Southern Quebec, as seen in Fig. 1 (MELCCFP, 2022b). The dataset is a reconstruction of past hydrology combining six simulations from the HYDROTEL hydrological model driven by gridded observational temperature and precipitation data. Furthermore, it incorporates observed streamflow measurements through an optimal interpolation technique to provide the most accurate estimate of past streamflow conditions across gauged and ungauged watersheds, accounting for both model and observational uncertainties (Lachance-Cloutier et al., 2017). Finally, to ensure robustness for ungauged river segments, HYDROTEL is calibrated using a regional calibration technique, identifying optimal parameter sets for each of the 6 simulations across 70 gauged watersheds.*

*HYDROTEL itself is a physically-based, semi-distributed hydrological model that is operationally employed by the Government of Quebec for streamflow forecasting (Fortin et al., 2001). Unlike conceptual models, which fully rely on empirical relationships and simplified assumptions to represent hydrological processes, HYDROTEL employs a more detailed set of physical equations to simulate various aspects of water movement and distribution within a watershed, such as infiltration, runoff, snowmelt or evapotranspiration. However some components, such as subsurface runoff, are still fairly simple in their representation. The model divides the*

*landscape into computational units called "Relatively Homogeneous Hydrological Units" (RHHUs), which are defined based on land use, soil type, and topographical characteristics such as slope and aspect. In the configuration used for the Atlas, the average size of RHHUs is 11 km2, forming a hydrological network comprising 28,035 river segments throughout Southern Quebec. Streamflow at each segment is calculated as the sum of upstream contributions and runoff/baseflow from neighboring RHHUs. Results are reported for 9,665 river segments, specifically those with watersheds larger than 50 km2 and not subject to significant anthropogenic influences, such as dams with daily or weekly management practices. Comprehensive details on model configuration, calibration procedures, and validation results are provided in MELCCFP (2024).*

*Although six distinct model versions exist, only the "MG24HQ" configuration of HYDROTEL was selected for the present study due to computational constraints within the experimental framework, as it showcases the best performance in simulating summer streamflow (MELCCFP, 2024)."*

Furthermore, Section 2.3 (Experimental framework) was modified to make it clear that temperature and precipitation were the only inputs from ClimEx used in HYDROTEL.

*"HYDROTEL requires daily surface temperature and precipitation data as input. These variables were taken from the CRCM5, then bias adjusted using a Detrended Quantile Mapping technique (DQM; Gennaretti et al. (2015))"*

**1-2. There is certain disconnection between the analytical work carried out to provide the results in terms of hydrological drought indices and the discussion part presented in section 4, especially sections 4.1 and 4.2. These sections mostly present information about drought impacts on ecosystems and socio-economic activities during the 2021 event. There are only some suggestions about how some of the calculated hydrological drought indices could be used to better assess drought impacts, but most of the content in these sections is a description of the impacts from 2021 drought. My suggestion would be to somewhat re-arrange the content in this article and to bring in the description of the 2021 drought in terms of impacts on ecosystems and socioeconomics much earlier (under 'Introduction', or under 'Methodology' as a separate section, or somehow combined with the section on 'Stakeholder consultation'). The argument can then be introduced that lack of adequate hydrological drought indices to assess future droughts under CC is an issue that this article addresses (There is even mentioning that the province of Quebec indeed does not have such indicators in line 60). The whole content of calculating these indices would then follow, and the Discussion section can be somewhat extended about the value / potential use of the proposed indicators. The need to move towards the next step, which is developing impact-related drought indicators could then be emphasized in that discussion. This is just a suggestion, and the authors may decide on a different approach to better connect the discussion part with the previous part of the article.**

Thank you for the constructive comment. We tried moving the description of the impacts of 2021 in the Introduction or Methodology section, but since many of the documented impacts came from the stakeholder consultation, it was difficult to make it fit that early in the text.

The following changes were made throughout the manuscript to respond to the reviewer's comment:

The first paragraph of 2.4.2 (Hydrological drought indicators) has been reworked to discuss the challenge of establishing a direct link between low streamflow values and their on-the-ground consequences, as proposed, as well as the objective behind the hydrological indicators.

*"Establishing a direct link between low streamflow values and their on-the-ground consequences is challenging, as impact-inducing thresholds are often unknown. Even when such thresholds exist in the literature or in regulations, they are not always well adapted to truly represent local conditions. The 7Q2 metric (2-year minimum of the 7-day average flow), for instance, plays a crucial role in Quebec for setting water withdrawals limits in a way that they maintain the health of river ecosystems. However, that metric has been criticized in recent years and deemed insufficient to truly represent the needs of ecosystems, with ongoing efforts to replace it with another indicator better suited to represent the geomorphological conditions of rivers and be more robust in the context of a changing climate (Berthot et al., 2020, 2021). Furthermore, for both ecosystems and human activities, water levels often hold greater relevance than streamflow; however, such data is rarely available, and especially not on a broad spatial scale. A storyline approach is valuable here, as the 2021 drought is known to have caused multiple negative impacts and surpassed key thresholds (see Sect. 3.1.1 and 3.1.2). Various hydrological indicators were thus used to characterize the exceptional intensity and duration of the 2021 drought (Table 1), as well as the future events, to ensure a consistent framework for assessing and comparing the potential impacts of climate change on hydrological droughts similar to 2021. The selected indicators capture key aspects of river conditions, enhancing our understanding of the extreme nature of the 2021 event across different facets of the hydrological cycle, such as the timing of the low-water season and the minimum summer flow."*

The text related to the impacts of 2021 has been shortened and moved into the first section of the results, since they are effectively results from the survey. They are now new subsections under 3.1 (Analysis of the 2021 drought in Southern Quebec).

Finally, the Discussion itself has also been heavily reworked to focus more on the potential added value of storylines as communication tools for stakeholders, using the drought indicators and results from this study to discuss expected future changes in extreme drought conditions.

*"There is no universally accepted indicator or threshold that indicates when water scarcity begins to significantly impact ecosystems, drinking water systems, and other human uses. Additionally, Quebec currently lacks a formal low-water alert system, such as the one in place in Ontario (Ontario Ministry of Natural Resources et al., 2010), which complicates the assessment of the effects of meteorological and hydrological droughts on human activities. The Ontarian system, which includes regulatory restrictions and actions for each alert level, provides a framework that would help assess the escalating impacts of a drought as it unfolds, as well as the water management strategies necessary to face such events. However, as previously discussed, the 2021 drought had numerous documented adverse effects on Southern Quebec. Although the hydrological indicators used in this study must be interpreted qualitatively regarding their causal relationship with observed impacts, the storyline lens still offers valuable insights into the potential severity of future extreme droughts. Our results indicate that, should a drought comparable to the 2021 event occur in the future — particularly under +3°C warming conditions — water scarcity would be considerably exacerbated. As noted earlier, with the exception of freshet volumes, all low-flow indicators suggest progressively worsening conditions under the projected climate scenarios. These changes could lead to substantial ecological degradation and a diminished capacity of ecosystems to provide critical services for human activities. Notably, despite the aforementioned criticisms of the 7Q2 as a measure of ecosystem needs (Berthot et al., 2020), the increasing number of days falling below this threshold raises significant concerns. Extended droughts, coupled with rising temperatures, will not only exacerbate water quality and availability issues but could also lead to irreversible changes to*

*aquatic ecosystems, thereby threatening biodiversity and the essential ecosystem services reliant on stable water conditions. This situation could further strain municipal water supplies, agriculture, and other vital sectors, as evidenced by the 2021 drought, which highlighted Quebec's vulnerability to the impacts of droughts.*

*In light of the risks posed by climate change and increasingly frequent drought events, the storyline approach offers a valuable framework for understanding, communicating, and preparing for future droughts in collaboration with stakeholders. Indeed, stakeholders possess intimate knowledge of their local water systems and retain memories of the real impacts of past events. Storylines foster dialogue between stakeholders and climate scientists by making climate data more accessible and relevant for decision-making, particularly for developing effective response measures. Unlike broad, abstract projections like "Future 100-year droughts will be n days longer," which can fail to provide concrete insights into the specifics of what a 100-year drought might entail, storylines offer more actionable and tangible scenarios. For example, a storyline based on our results might specify: "An event analogous to the 2021 drought, but occurring under a +3°C global temperature increase, could last 30 days longer and result in a 50% reduction in the minimum seven-day average streamflow observed in 2021 for rivers located south of the St. Lawrence River" This context-rich and event-driven approach enables stakeholders to identify vulnerabilities based on 2021 and adapt their systems by extrapolating potential future conditions. For instance, if a municipality was just one week away from experiencing a water shortage during the 2021 drought, understanding that a future event of similar magnitude might last a month longer, thus surpassing the capacity of their system, may prompt proactive climate adaptation planning such as bolstering water conservation measures, investing in infrastructure improvements, or exploring alternative water sourcing strategies.*

*Additionally, the storyline method might encourage a more collaborative and informed dialogue among stakeholders, enabling better coordination across sectors such as water management, urban planning, agriculture, and public health by grounding future projections in recent, real-world examples. A second phase is planned for mid-2025 and will focus on further investigating how storylines can deliver actionable insights through close collaboration with stakeholders in select watersheds that experienced significant impacts from the 2021 drought."*

**Specific comments:**

**1-3. Line 5, in the abstract: The statement "This approach allowed for enhanced collaboration with water management experts and other stakeholders to project the possible impacts of climate change on serious water deficits in Quebec." should be modified or removed. There is no evidence presented in the article that this actually occurred. (This comment is somewhat related to the general comment 2, mentioned above).**

This sentence has been changed.

*"Storyline approaches offer potential as a communication tool to foster collaboration with water management experts and other stakeholders, particularly in cases where impact-inducing thresholds are not well understood. By linking future river conditions to a real event that affected Southern Quebec, these approaches may help facilitate discussions on the potential impacts of climate change on water deficits in the region."*

**1-4. Please insert somewhere in Section 1 (Introduction) or Section 2 (Methodology) a figure with an overview map of the actual study area, together with geographical features to which you are later on**

referring (e.g. St. Lawrence river / valley, Ottawa river, Lake of Two Mountains, etc.). This will be very useful for readers unfamiliar with the geography of Quebec.

Figure 1 was updated to make the overview map of the HYDROTEL network larger. In addition, key geographical features for the paper have been added to it.

**1-5. Please make it clear in Section 1 (Introduction) that the focus of this research is hydrological drought (and the indices). When one reads this section, it is not always clear whether meteorological or hydrological drought will be the target of the analysis (later on it becomes clear).**

The following was added to last paragraph of the Introduction to be more explicit:

*"We use a SBA to first identify adequate analogues of the meteorological drought using a large ensemble of regional climate simulations. We then investigate the associated hydrological drought by also making use of a hydrological model to project streamflow data during the drought event, thus also contrasting present and future hydrological conditions."*

**1-6. Line 60: Please add "and ecosystems" at the end of the sentence "…significantly impact human activities", to be consistent with the content presented later.**

This has been changed accordingly.

**1-7. Line 85: Please do not use 'water deficits', as it adds to the ambiguity regarding what is the focus of the cited study and of this study. The cited study is clearly about precipitation deficits (meteorological drought) and this study goes further in calculating hydrological drought indices based on streamflow data.**

This has been changed to 'precipitation deficits'.

**1-8. Line 95: Please use 'Sect.' or 'Section' consistently.**

That sentence has been changed to circumvent the issue.

*"The data used and the experimental framework are described in Sect. 2, while Sect. 3 presents the 2021 drought [...]"*

**1-9. Line 115: The statement "By combining the qualitative data from the questionnaire with the quantitative insights drawn from the storylines, it becomes possible to obtain a more comprehensive understanding of the potential impacts of future severe droughts" should be modified or removed. There is no evidence in the content of the article that such combination was performed. If it is a suggestion that this should be done in the future – the statement should be re-formulated. (This comment is related to the general comment 2, and specific comment 1, mentioned above).**

That sentence has been removed.

**1-10. Line 125: the sentence starting with "The retrospective analysis…" is not clear. Could you please expand it or re-formulate it? What is meant by 'combining streamflow measurements from various locations'? What was combined and how? Which 'gridded observation data were used for driving HYDROTEL' (again, see general comment 1)?**

As mentioned in our response to Comment #1-1, Section 2.2 (Observational data) has been substantially reworked. Specifically regarding this comment, the text is now more descriptive with regards to the optimal interpolation method used to recreate past hydrological conditions.

*"Historical streamflow data were obtained from the Hydroclimatological Atlas of Southern Quebec, which covers the period from 1970 to the present and spans the entire region of Southern Quebec, as seen in Fig. 1 (MELCCFP, 2022b). The dataset is a reconstruction of past hydrology combining six simulations from the HYDROTEL hydrological model driven by gridded observational temperature and precipitation data. Furthermore, it incorporates observed streamflow measurements through an optimal interpolation technique to provide the most accurate estimate of past streamflow conditions across gauged and ungauged watersheds, accounting for both model and observational uncertainties (Lachance-Cloutier et al., 2017)."*

**1-11. Line 150: The statement "… and, importantly for this work, potential evapotranspiration (PET)" is a bit surprising. Practically all hydrological models use PET as an input, so how can this be a particular feature of HYDROTEL that is important?**

The intended meaning was that HYDROTEL computes its own PET and, therefore, that the choice of equation could impact the results (although, as stated later in the text, it does not).

Section 2.2 (Observational data) now introduces the physically-based nature of HYDROTEL and the fact that it computes its own evapotranspiration based on meteorological inputs. Furthermore, the statement mentioned in the comment has been removed, while the text at L.220 has been reworked:

*"Access to potential evapotranspiration (PET) data is required for calculating the water budget needed for the SPEI. As a physically-based hydrological model, HYDROTEL computes its own PET based on the provided temperature data. The model configuration used for this study employed the PET definition provided in McGuinness and Bordne (1972), which takes into account both temperature and latitude. Therefore, this definition was also chosen for the computation of SPEIs in order to have a consistent definition between the meteorological and hydrological droughts."*

**1-12. Line 175: This approach is not entirely clear to me. The SPEI values will be comparable in terms of values, but if they come from two different distributions (from two different climates), how can then they be comparable? Can you please elaborate a bit?**

Additional justifications have been added at L.215.

*"SBA storylines are constructed by identifying analogues of a past high-impact event within a large ensemble of simulated climate data. To facilitate this, it was crucial to ensure that SPEI distributions could be compared across the reference (observed) climate, ClimEx during the historical period, and future ClimEx simulations. To address this, the three-parameter log-logistic distribution was calibrated independently for each realization and time period (historical and future), ensuring consistent SPEI value ranges. While preferable for traditional climate change projections, a single calibration of SPEIs during the historical period only would have complicated finding relevant future events, as those with similar SPEIs to the past would offer little new insight — an issue also noted by van der Wiel et al. (2021) with the M3 metric —, while those with lower SPEIs would be more difficult to relate to the 2021 drought. Using separate calibrations for the past and future enables the identification of events with a similar probability and annual progression as 2021, but relative to their respective climatologies."*

**1-13. Line 225: So, there were 10 best analogues for the historical period *and* 10 best analogs for the future, and the difference between the averages of the two was calculated, right? Please clarify.**

Following a suggestion by Referee #3, that paragraph has been moved to L.290, as it fits better there with the new structure of the text.

Furthermore, that part of the methodology has been clarified by adding the previous Figure S2 to the main text and reworking the sentence describing the construction of future analogs.

*"[...] future hydrological indicators for the historical event under +2 and +3°C scenarios were constructed by first calculating the indicators for the 10 best analogues, averaging them over the future and historical time periods separately, and applying that difference to the observed indicator (Fig. 3)."*

**1-14. Lines 270 and 280: Would the results be sensitive to these weights values? Has this been tested? If not, perhaps it should be recommended?**

As mentioned in our pre-response to this comment, multiple tests were indeed conducted on the weighting scheme. Our results show that while the selected events themselves are sensitive to the weights and that the outcomes are thus not robust with a limited selection (e.g., 3 best analogues), the results are stable and robust when 10 or more analogues are used. This has been clarified at L.300.

*"Additional verifications have shown that while the selection of the best analogues is sensitive to the weighting scheme, and that the average of the best n analogues thus becomes unstable when the number is too low, the results were robust and stable with 10 selections (not shown)"*

**1-15. Section 3.4 (Limitations). Perhaps it could be recommended to carry out similar analysis for individual basins, and not only for such a large area? This would bring the use of the indicators closer to actual water resources planning and management, in my view…**

Given the distributed nature of HYDROTEL, this is not really a limitation of our study. Instead, this is mentioned as a perspective for future works, which are already planned. This was added both to the end of the Discussion at L.470 and to the Conclusions at L.525.

*"Additionally, the storyline method might encourage a more collaborative and informed dialogue among stakeholders, enabling better coordination across sectors such as water management, urban planning, agriculture, and public health by grounding future projections in recent, real-world examples. A second phase is planned for mid-2025 and will focus on further investigating how storylines can deliver actionable insights through close collaboration with stakeholders in select watersheds that experienced significant impacts from the 2021 drought."*

*"This work aimed to improve our understanding of future drought conditions at the provincial scale, but the distributed nature of HYDROTEL means that hydrological indicators were nonetheless computed at a local scale. The current study thus lays the foundation for future work, particularly through stakeholder engagement and local-scale impact assessments. A second phase of the project, planned for mid-2025, will focus on deepening these efforts by working closely with stakeholders in a few of the watersheds most affected by the 2021 drought to further explore how storylines can provide actionable insights."*

**1-16. Figure 7. I don't see why this figure is necessary. Many drought impacts have been discussed, and not supported by similar generic information (and figures). Why is this generic view on drought impacts on river**

systems chosen to be presented? It is a bit of a distraction, in my view. I would suggest to present relevant figures for this particular research (perhaps Figure S2 from the supplementary material?)

Following the changes to the paper, that figure was indeed removed. Figure S2 has been added to the Methodology section as Figure 3.

**Referee #2**

**General comments:**

**2-1. Bias-adjustment of climate model: The authors mention that the detrended quantile mapping bias adjustment technique was applied for precipitation. It isn't clear whether the bias adjustment was applied to each ensemble member independently from each other? It is recommended that any parameters for bias adjustment applied to single-model-initial-condition large ensembles should be computed from the pooled ensemble rather than individual ensemble members to preserve the range of internal variability in the original ensemble. It would be good if the authors can clarify their approach.**

The 50 members were indeed pooled together. That paragraph was reworked to give much more details on the bias adjustment step of the methodology.

*"HYDROTEL requires daily surface temperature and precipitation data as input. These variables were taken from the CRCM5, then bias adjusted using a Detrended Quantile Mapping technique (DQM; Gennaretti et al. (2015)). DQM is a variant of quantile mapping that first removes the trend in the data, typically through a polynomial fit (in this case, a fourth-degree polynomial), after which a correction is applied to the residuals. Specifically, the residuals are sorted into bins based on their empirical quantiles, and a transfer function is trained on the differences between each bin of the simulated and the reference datasets during a common time period. This transfer function is then used to correct the whole simulation. In this study, DQM was applied to each day of the year and each grid cell individually, using a ±15-day window around each day and separating data into 50 bins. The reference dataset used for the correction was the NRCANmet dataset, a spatially interpolated kriging of station data on a 10 km grid that covers all of Canada, produced by Natural Resources Canada (Hutchinson et al., 2009). Since ClimEx is a single model large ensemble, all 50 simulations were pooled together to construct the transfer function. Finally, for precipitation, the adjustment method was modified to combine DQM with a Generalized Pareto distribution in the upper tail of the distribution (Roy et al., 2023). This modification was added in order to improve the correction of the most extreme precipitation events compared to using a standard DQM technique, but should not play a role in the results presented here, as we focus on dry conditions."*

**2-2. Hydrological model: more details of the hydrological model would be appreciated. For example, was the model calibrated to observed streamflow in any way and what is the spatial resolution of the hydrological model? Were there any statistical downscaling methods applied to further downscale the large ensemble data prior to hydrological modelling? Related to this, would the authors be able to provide some indication of model performance compared to observed flows at gauges (e.g. in the supplement)? There should also be some discussion of the simulated river flows driven by ClimEx over the historical period – do they exhibit similar hydrolgical behaviour to observed river flows and where does observed flows lie within the wider range of ClimEx simulated flows?**

Section 2.2 (Observational data) has been substantially reworked to include more information on HYDROTEL and the configuration used in this study. See our response to Comment #1-1.

A comparison of simulated river flows driven by ClimEx over the historical period for a few key watersheds heavily affected by the 2021 drought has been added to the Supplementary Materials as Fig. S1.

**2-3. SPEI calculation: Have the authors looked at how their results might change if the SPEI distribution for the future periods are fitted to the parameters calculated from the historical period? This could serve as a good sensitivity test for whether the decision to calculate SPEI separately for historical and future period is a valid approach.**

This has been addressed in the revised paper. See our response to Comment #1-12.

**2-4. Changes in drought analogues: as the authors note, there are interesting hydrological dynamics within the future analogues (such as changes in winter thaw and changes in the timing of low flow season) which deserves a bit more discussion. Could the authors provide a time series of P, PET and simulated flows for the baseline observed 2021 and for the top 10 future drought analogues (either at particular selected catchments or averaged)? The readers can then visualise these temporal changes easier.**

A comparison of monthly mean anomalies in precipitation, evapotranspiration, and streamflow for 2021 and for the the average of the 10 best ClimEx analogues during the historical and future periods, for a few key watersheds heavily affected by the 2021 drought, has been added to the Supplementary Materials as Fig. S3 and S4.

**2-5. Discussion: The current Discussion needs to be reframed. The section seems out of place and reads more like a separate literature review rather than an actual discussion of the results. Perhaps the authors could reframe this to link to the results of the paper (e.g. how these storylines can be used to further drive water quality models or how flow indicators from the drought analogues could be used to infer different drought impacts). Much of the socio-economic impacts of the 2021 drought could actually go in the introduction to highlight the severity of the event and motivate why the authors decided to choose the 2021 event as a case study. There should also be further discussion of the utility of the storyline approach and future work (e.g. are there limitations to the storyline approach? How can such an approach be used alongside other climate change projection products?)**

This comment and the proposed solution align with Comment #1-2. See our response there.

**Minor amendments:**

**2-6. IPCC (2023) should be cited according to the official suggested citation according to the IPCC:**

> Lee, J.-Y., J. Marotzke, G. Bala, L. Cao, S. Corti, J.P. Dunne, F. Engelbrecht, E. Fischer, J.C. Fyfe, C. Jones, A. Maycock, J. Mutemi, O. Ndiaye, S. Panickal, and T. Zhou, 2021: Future Global Climate: Scenario-Based Projections and NearTerm Information. In Climate Change 2021: The Physical Science Basis. Contribution of Working Group I to the Sixth Assessment Report of the Intergovernmental Panel on Climate Change [Masson-Delmotte, V., P. Zhai, A. Pirani, S.L. Connors, C. Péan, S. Berger, N. Caud, Y. Chen, L. Goldfarb, M.I. Gomis, M. Huang, K. Leitzell, E. Lonnoy, J.B.R. Matthews, T.K. Maycock, T. Waterfield, O. Yelekçi, R. Yu, and B. Zhou (eds.)]. Cambridge University Press, Cambridge, United Kingdom and New York, NY, USA, pp. 553–672, doi:10.1017/9781009157896.006.

This has been changed.

**2-7. L80 – The authors may consider expanding the literature search and also refer to more recent studies which have also used an analogue approach to sample for drought/heatwave storylines – The following three studies may be useful, the first two for meteorological droughts and the third for hydrological droughts:**

> Faranda, D., Pascale, S., and Bulut, B.: Persistent anticyclonic conditions and climate change exacerbated the exceptional 2022 European-Mediterranean drought, Environ. Res. Lett., https://doi.org/10.1088/1748-9326/acbc37, 2023.

> Liao, Z., An, N., Chen, Y., and Zhai, P.: On the possibility of the 2022-like spatio-temporally compounding event across the Yangtze River Valley, Environ. Res. Lett., 19, 014063, https://doi.org/10.1088/1748-9326/ad178e, 2024.

> Chan, W. C. H., Arnell, N. W., Darch, G., Facer-Childs, K., Shepherd, T. G., Tanguy, M., and van der Wiel, K.: Current and future risk of unprecedented hydrological droughts in Great Britain, Journal of Hydrology, 130074, https://doi.org/10.1016/j.jhydrol.2023.130074, 2023.

Thank you for the references. They have been added to the introduction.

**2-8. L108 and elsewhere – there are several mentions of a questionnaire that was given to stakeholders to gather information on the 2021 drought and to disseminate results. More discussion of what the outcomes from this questionnaire should be given. Could the questionnaire be published in the supplement as well for transparency?**

The questionnaire was added to the Supplements, as requested.

**2-9. L155 – what is MG24HQ?**

This specific mention of MG24HQ has been removed, but more details are now given in the reworked Section 2.2. Specifically:

*"Although six distinct model versions exist, only the "MG24HQ" configuration of HYDROTEL was selected for the present study due to computational constraints within the experimental framework, as it showcases the best performance in simulating summer streamflow (MELCCFP, 2024)."*

**2-10. Font sizes in all figures could be bigger. Consider showing a subset of the results and putting other sub-plots in the supplementary materials. For example, Figure 3 could be simplified by including SPEI-3 averaged over May-Oct rather than each month separately (the monthly plots could go in the supplementary materials).**

Font sizes have been increased for all figures. However, we decided against showing only a subset of results, as it added complexity to the text and forced the reader to go back and forth between the text and Supplementary Materials.

**2-11. The Limitations section should go within the Discussion section rather than Results.**

This section was placed in the Results due to the nature of the previous Discussion section. However, with the proposed restructuring, this has been moved as suggested.

**Referee #3**

**General comments:**

**3-1. The appropriateness of the approach is claimed, but hardly explained and/or compared with other approaches. Why can the extensive modelling approach add to the baseline method? Why would it be needed (or even allowed) to keep the storyline approach when running such models? The discussion might have revealed some of these issues, but paragraph 4 does not really engage with the climate stories/narratives as such.**

The Introduction, Methodology, and Discussion have been reworked to put more emphasis on the potential added value of the storyline approach, also in line with Comments #1-2 and #2-5.

Specifically, the Introduction now better contrasts traditional climate scenarios and storylines at L.70:

*"Traditional climate change projections evaluate changes in extreme events by examining the potential shifts in the frequency or intensity of events with a specified magnitude or probability of occurrence, such as a 100-year drought. These projections often utilize multi-model ensembles, where changes are assessed through the ensemble median and/or the likelihood of increase or decrease across the ensemble for future scenarios. In contrast, climate storylines offer a way to simplify complex climate information by linking it to stakeholders' past experiences and projecting how specific events may evolve in the future. For example, van der Wiel et al. (2021) focused on the 2018 European drought and projected this high-impact event under future climatic conditions in a physically consistent manner to assess how that one event might change if it were to occur again in the future. Consequently, the storyline approach focuses on the plausibility of the future scenario rather than its likelihood, and can avoid the need to pinpoint specific damage-inducing thresholds by constructing the future scenario based on an event that already caused known damages in the past. Storylines have been applied in both climatological and hydrological research to effectively illustrate future extreme events such as droughts, floods and storms (Schaller et al., 2020; van der Wiel et al., 2021; Chan et al., 2022; Gessner et al., 2022; Chan et al., 2023; Liao et al., 2024)"*

The following paragraph is now also more explicit on how storylines are built in other studies:

*"Storylines can be constructed using various methodologies (Fossa Riglos et al., 2024). Physical climate storylines (PCS) involve using a climate or weather forecast model to recreate a singular high-impact event and explore alternative realities, for example by modifying the initial conditions or by using a pseudo-global-warming approach (e.g., Hazeleger et al., 2015; Gessner et al., 2022; Matte et al., 2022). Scenario-based approaches (SBA), on the other hand, do not recreate the event, but instead search for adequate analogues within a simulated dataset that includes both historical and future climate conditions (e.g., van der Wiel et al., 2021; Faranda et al., 2023; Liao et al., 2024). This approach thus involves defining the event of interest using metrics that can accurately capture its intensity and main characteristics, while also being flexible and reliable enough to identify comparable events in the simulated data. For example, in their study related to a severe drought in Europe, van der Wiel et al. (2021) examined three different types of metrics associated with precipitation deficits: the average deficit during the most severe drought months (M1), the regression slope of the deficits in the months preceding the peak deficit (M2) and a temporal regression of the deficit time series (M3). Alternatively, rather than focusing on specific historical events, SBAs may involve searching large*

*ensembles for unprecedented climate conditions, such as multiple consecutive dry seasons, to identify worst-case scenarios which can then be used for stress-testing systems (Chan et al., 2023)."*

Section 2.4.2 (Hydrological drought indicators) now also emphasises the potential benefits of the storyline approach for droughts:

*"Establishing a direct link between low streamflow values and their on-the-ground consequences is challenging, as impact-inducing thresholds are often unknown. Even when such thresholds exist in the literature or in regulations, they are not always well adapted to truly represent local conditions. The 7Q2 metric (2-year minimum of the 7-day average flow), for instance, plays a crucial role in Quebec for setting water withdrawals limits in a way that they maintain the health of river ecosystems. However, that metric has been criticized in recent years and deemed insufficient to truly represent the needs of ecosystems, with ongoing efforts to replace it with another indicator better suited to represent the geomorphological conditions of rivers and be more robust in the context of a changing climate (Berthot et al., 2020, 2021). Furthermore, for both ecosystems and human activities, water levels often hold greater relevance than streamflow; however, such data is rarely available, and especially not on a broad spatial scale. A storyline approach is valuable here, as the 2021 drought is known to have caused multiple negative impacts and surpassed key thresholds (see Sect. 3.1.1 and 3.1.2). Various hydrological indicators were thus used to characterize the exceptional intensity and duration of the 2021 drought (Table 1), as well as the future events, to ensure a consistent framework for assessing and comparing the potential impacts of climate change on hydrological droughts similar to 2021. The selected indicators capture key aspects of river conditions, enhancing our understanding of the extreme nature of the 2021 event across different facets of the hydrological cycle, such as the timing of the low-water season and the minimum summer flow."*

Finally, the Discussion has been fully rewritten to focus on the potential of storylines are communication tools, especially in cases like hydrological droughts where specific damage-inducing thresholds are not known. Specifically:

*"There is no universally accepted indicator or threshold that indicates when water scarcity begins to significantly impact ecosystems, drinking water systems, and other human uses. Additionally, Quebec currently lacks a formal low-water alert system, such as the one in place in Ontario (Ontario Ministry of Natural Resources et al., 2010), which complicates the assessment of the effects of meteorological and hydrological droughts on human activities. The Ontarian system, which includes regulatory restrictions and actions for each alert level, provides a framework that would help assess the escalating impacts of a drought as it unfolds, as well as the water management strategies necessary to face such events. However, as previously discussed, the 2021 drought had numerous documented adverse effects on Southern Quebec. Although the hydrological indicators used in this study must be interpreted qualitatively regarding their causal relationship with observed impacts, the storyline lens still offers valuable insights into the potential severity of future extreme droughts. Our results indicate that, should a drought comparable to the 2021 event occur in the future — particularly under +3°C warming conditions — water scarcity would be considerably exacerbated. As noted earlier, with the exception of freshet volumes, all low-flow indicators suggest progressively worsening conditions under the projected climate scenarios. These changes could lead to substantial ecological degradation and a diminished capacity of ecosystems to provide critical services for human activities. Notably, despite the aforementioned*

*criticisms of the 7Q2 as a measure of ecosystem needs (Berthot et al., 2020), the increasing number of days falling below this threshold raises significant concerns. Extended droughts, coupled with rising temperatures, will not only exacerbate water quality and availability issues but could also lead to irreversible changes to aquatic ecosystems, thereby threatening biodiversity and the essential ecosystem services reliant on stable water conditions. This situation could further strain municipal water supplies,agriculture, and other vital sectors, as evidenced by the 2021 drought, which highlighted Quebec's vulnerability to the impacts of droughts.*

*In light of the risks posed by climate change and increasingly frequent drought events, the storyline approach offers a valuable framework for understanding, communicating, and preparing for future droughts in collaboration with stakeholders. Indeed, stakeholders possess intimate knowledge of their local water systems and retain memories of the real impacts of past events. Storylines foster dialogue between stakeholders and climate scientists by making climate data more accessible and relevant for decision-making, particularly for developing effective response measures. Unlike broad, abstract projections like "Future 100-year droughts will be n days longer," which can fail to provide concrete insights into the specifics of what a 100-year drought might entail, storylines offer more actionable and tangible scenarios. For example, a storyline based on our results might specify: "An event analogous to the 2021 drought, but occurring under a +3°C global temperature increase, could last 30 days longer and result in a 50% reduction in the minimum seven-day average streamflow observed in 2021 for rivers located south of the St. Lawrence River" This context-rich and event-driven approach enables stakeholders to identify vulnerabilities based on 2021 and adapt their systems by extrapolating potential future conditions. For instance, if a municipality was just one week away from experiencing a water shortage during the 2021 drought, understanding that a future event of similar magnitude might last a month longer, thus surpassing the capacity of their system, may prompt proactive climate adaptation planning such as bolstering water conservation measures, investing in infrastructure improvements, or exploring alternative water sourcing strategies.*

*Additionally, the storyline method might encourage a more collaborative and informed dialogue among stakeholders, enabling better coordination across sectors such as water management, urban planning, agriculture, and public health by grounding future projections in recent, real-world examples. A second phase is planned for mid-2025 and will focus on further investigating how storylines can deliver actionable insights through close collaboration with stakeholders in select watersheds that experienced significant impacts from the 2021 drought."*

**3-2. The text actually reads much like a rather traditional text on climate scenarios. It might be that the use of the model enforces this, but I found it a little tricky to find the added value of storylines - or even the difference with other ways of sharing scenarios. I would think that the text could be much the same without ever mentioning the term "storyline analysis".**

See the response above. We hope that the revised version of our manuscript help betetr understand the difference between climate storylines and other climate scenarios.

**3-3. The method section seems to be incomplete. First, the method of storyline analytics is not discussed. How does one do it? Second, in the results section there seem to be several series of steps included that might need to be included in the method section.**

The "Present and future analogues in the ClimEx Large Ensemble" section, which described specifically how the future analogues were constructed, has been renamed "Construction of the analogues" and moved to the Methodology, as suggested. Furthermore, Figure S2 has been moved to the main body of the text (now Fig. 3) to better illustrate how the future indicators are built.

**3-4. The usefulness of the approach is defended by claiming that the results can be used in discussions with stakeholders, with some details on stakeholders ideas in section 4 (which for some reason is presented as discussion and not as results). Any evidence for that claim is not offered, as the stakeholder data come from a survey that was done before the storyline process. As such, we read two different parts: one on climate scenarios and one on survey results. I could also imagine that in a dialogue with stakeholders, modelling scenarios, whether they are shared in storyline format or otherwise, might change the stories themselves. This can only be checked after engaging with the stakeholders. I would be very interested in that aspect.**

We are sorry about the confusion on the stakeholder's consultation. As suggested, the documented impacts of 2021 have been moved to the Results as subsections under Sect. 3.1 (Analysis of the 2021 drought in Southern Quebec). The Discussion has been mostly rewritten and now puts more emphasis on the potential uses of our results and of the storyline approach. While multiple stakeholders were engaged throughout our work, specific testing for the added communication value of storylines is being planned for 2025. This information was added both to the end of the Discussion at L.470 and to the Conclusions at L.525.

*"Additionally, the storyline method might encourage a more collaborative and informed dialogue among stakeholders, enabling better coordination across sectors such as water management, urban planning, agriculture, and public health by grounding future projections in recent, real-world examples. A second phase is planned for mid-2025 and will focus on further investigating how storylines can deliver actionable insights through close collaboration with stakeholders in select watersheds that experienced significant impacts from the 2021 drought."*

*"This work aimed to improve our understanding of future drought conditions at the provincial scale, but the distributed nature of HYDROTEL means that hydrological indicators were nonetheless computed at a local scale. The current study thus lays the foundation for future work, particularly through stakeholder engagement and local-scale impact assessments. A second phase of the project, planned for mid-2025, will focus on deepening these efforts by working closely with stakeholders in a few of the watersheds most affected by the 2021 drought to further explore how storylines can provide actionable insights."*

**3-5. In conclusion, I would think that this text is a little unbalanced in terms of explaining the usefulness of the method, in terms of added value of the modelling, and in terms of the rather different section 3 and 4. Section 4 does seem to miss the point on discussing results from Section 3 anyway. The storyline itself reads like a standard climate scenario explanation, which is useful, but NOT necessarily new enough to be published.**

We hope that the restructuring of the paper have described in our previous responses improve the flow of the text and reduce the previous disconnect between Sections 3 and 4.

**Specific comments (from the PDF file):**

L.5

This sentence has been changed.

*"Storyline approaches offer potential as a communication tool to foster collaboration with water management experts and other stakeholders, particularly in cases where impact-inducing thresholds are not well understood. By linking future river conditions to a real event that affected Southern Quebec, these approaches may help facilitate discussions on the potential impacts of climate change on water deficits in the region."*

L.25

This has been rephrased.

*"Hydrological droughts in colder regions are complex phenomena to analyze due to the added interplay of rainfall and snow-driven recharge."*

L.70

Refer to the answer to Comment #3-1.

L.125

That sentence has been reworked. While this does exclude several rivers, these are typically larger rivers with higher mean streamflow, making problematic low water levels less likely to occur (though not impossible). In any case, these rivers had to be excluded, as HYDROTEL is currently unable to account for them.

*"Results are reported for 9,665 river segments, specifically those with watersheds larger than 50 km² and not subject to significant anthropogenic influences, such as dams with daily or weekly management practices."*

L.130

The term "significant number" has been changed for "large number". It was used to indicate that a large number of simulations are needed (the higher, the better), rather than in its statistical sense.

L.185

We hope that the changes explained in our responses to Comments #3-1 and #3-3 make the methodology more understandable.

L.470

That sentence has been reworked.

*"In the absence of clear impact-inducing thresholds, but with documented impacts in 2021, these storylines can provide a framework for stakeholders to gain a clearer understanding of future droughts by linking them to an event that they are already familiar with."*

---

## Author Response (AR2)

We would like to thank the referee for their constructive feedback. We appreciate the time and effort that was put into the review and we hope that our responses prove satisfactory. For clarity, the reviewer's comments are presented in black, with our responses in red.

Best regards,

Gabriel Rondeau-Genesse, on behalf of all authors.

**Referee #3**

This new version of the text reads well in terms of general issues of climate change, the importance of droughts including the slower processes associated with them, and the construction of scenarios that are able to capture such slower processes.

And yet, I remain really hesitant about the text when it comes to what is mentioned as central to it: the story line ideas. This sentence on page 20, so already well into the Discussion, made me doubt: "In light of the risks posed by climate change and increasingly frequent drought events, the storyline approach offers a valuable framework for understanding, communicating, and preparing for future droughts in collaboration with stakeholders." This may be something that can be accepted as a claim, but the paper does not prove that - as the scenarios were not used to consult with stakeholders.

A reference to Caviedes-Voullième and Shepherd (2023) has been added to this statement, as it aligns with the points made in their paper. Furthermore, both this statement and our manuscript remain consistent with the broader literature on climate storylines already cited, in which stakeholder engagement is typically discussed separately.

However, we have made it clearer in the Introduction that this paper's main goal is to present the climate storyline itself – as in the projection of the 2021 Quebec drought into future climate conditions – and that it aims to provide the foundation for the second phase of the project, which will be the one involving stakeholder engagement. As detailed in another response below, the planning and organisation of these workshops are already well underway.

L.95: *The primary objective of this study is to apply a storyline approach to generate plausible, physically coherent projections of the 2021 Quebec drought under future climate conditions. Distinguishing itself from most existing research, the study also incorporates a distributed hydrological model to simulate hydrological conditions during those events at both provincial and local scales. These storylines will form the basis for stakeholder engagement in the project's second phase, designed to foster dialogue on the potential impacts of future water scarcity in southern Quebec and to explore adaptive management strategies.*

Perhaps the text wanted to show how story lines could be constructed. That may be a fair goal, but in that case I would need to know how what the text does is different from other scenario-based approaches - which is not discussed. How is a "story line" different from "a certain type of scenario"? I might be missing something here, as I find it really difficult to see what the term "story line" adds to the paper as is. For example, part 3 (Results) does not use the term "story line " at all. I find this a little strange, as I would think that results of a text suggesting that story lines are important would have to show actual story lines in those results.

The term *climate storyline* is well-established in the scientific literature and is referenced multiple times in the IPCC AR6 report. Notably, WG1 Chapter 10 (Box 10.2: *Storylines for Constructing and Communicating Regional Climate Information*) provides a detailed description of the approach. One type of climate storyline, which we use, involves modifying a historical event to simulate how it might unfold under future climate conditions, in a physically coherent manner. This methodology and concept is thus central to our study, even when it was not explicitly named such as in Section 3.2 of the Results.

We have refined the relevant paragraph in the Introduction to further clarify the distinction between multi-model ensemble approaches and storylines:

L.65: *Recently, storyline approaches have begun to be explored in climate change research and may provide a robust framework for projecting the impacts of future extreme events in the absence of clear thresholds, due to their event-based nature (Shepherd et al., 2018; Shepherd, 2019; Sillmann et al., 2021; Matte et al., 2022; Baulenas et al., 2023; Caviedes-Voullième and Shepherd, 2023). Traditional climate change projections evaluate changes in extreme events by examining the potential shifts in the frequency or intensity of events with a specified magnitude or probability of occurrence, such as a 100-year drought. These projections rely on multi-model ensembles, where results from multiple climate models are combined together to provide a robust climate change signal through ensemble percentiles, as well as a likelihood of increase or decrease using the level of agreement between the climate change signal of the individual models. This statistical framework generally does not try to link indicators (e.g. the 100-year drought) directly to specific past events. In contrast, climate storylines can use climate data to construct plausible future scenarios by modifying real historical events, without having to explicitly specify their likelihood. This approach offers a way to simplify complex climate information by projecting how specific events may unfold in the future, thus linking climate information to familiar stakeholders experiences. It also potentially avoids the need to pinpoint specific damage-inducing thresholds by constructing the future scenario based on an event that already caused known damages in the past. Storylines have been applied in both climatological and hydrological research to effectively illustrate future extreme events such as droughts, floods and storms (Schaller et al., 2020; van der Wiel et al., 2021; Chan et al., 2022; Gessner et al., 2022; Chan et al., 2023; Liao et al., 2024).*

In addition, to enhance clarity and consistency, the term *storyline* has been explicitly incorporated throughout the text where relevant, as well as into key sections of the manuscript. Specifically:

- Section 2.5, previously titled *Construction of the analogues*, is now *Construction of the storylines*. The

- Section 3.2, previously titled *The 2021 drought under future climatic conditions*, is now *Storylines of the 2021 drought under future climate conditions*

- The opening of Section 3.2 now also reiterates the methodology as follows:

L.390: *Storylines of the 2021 drought under +2°C and +3°C warming conditions were constructed by modifying the climatological and hydrological indicators computed for the 2021 drought using the analogues found in the ClimEx dataset, as shown in Fig. 3.*

On page 21, we read that "it is essential to note that the primary objective of this study was not to achieve exact modelling of future low flows, but to produce plausible, physically coherent low-flow indicators reminiscent of a recent event to stimulate conversations with stakeholders about the effects of water scarcity

in future climates in southern Quebec." We simply cannot know whether the scenarios were stimulating discussions, as there was no stakeholder discussion.

We read as well that "The constructed scenarios offer valuable insights into potential future conditions and emphasize the importance of preparedness for water shortages." We cannot know this either as the scenarios have not been tested in practices. The authors may find the scenarios insightful, and they are obviously allowed to defend that position, but that does not make what they did a story line approach.

These statements, and others that were similar, have been adjusted to reflect the fact that stakeholder engagement will be performed in the 2nd phase of the project.

L.10: *By linking future river conditions to a real event that affected Southern Quebec, storyline approaches have the potential to facilitate discussions on the impacts of climate change on water deficits in the region, particularly in cases where those impact-inducing thresholds are not well understood.*

L.470: *Storylines can foster dialogue between local actors and climate scientists by making climate data more accessible and relevant for decision-making, particularly for developing effective response measures.*

L.495: *While stakeholder participation was not carried out in this initial study, we contend that the storyline method supports a more collaborative and informed dialogue between stakeholders, allowing better coordination across sectors such as water management, urban planning, agriculture, and public health by grounding future projections in recent, real-world examples. To further evaluate this potential, the second phase, planned for mid-2025, will investigate how storylines can deliver actionable insights through workshops organized with stakeholders in select watersheds that experienced significant impacts from the 2021 drought*

L.515: *However, it is essential to note that the primary objective of this study was not to achieve exact modelling of future low flows, but to produce plausible, physically coherent low-flow indicators reminiscent of the 2021 drought, so that they could be used to stimulate conversations with stakeholders about the effects of water scarcity in future climates in southern Quebec. The constructed scenarios can offer valuable insights into potential future conditions and emphasize the importance of preparedness for water shortages.*

Perhaps the text focuses on how to create scenarios for droughts, which is perhaps meant with "plausible, physically coherent low-flow indicators reminiscent of a recent event". That focus might be enough, but is less wide ranging than a discussion on the concept of "story lines" in discussion with stakeholders. There are not too many stakeholders in the text...

We hope that the previous answers already contextualize our manuscript with the broader literature on climate storylines, where the development of the storyline is typically treated separately from stakeholder engagement. However, as the planning for the stakeholder workshops is already well underway, the discussion section has been updated to include a more detailed and illustrative example of how our results could be used to construct a local-scale storyline:

L470: *Unlike broad, abstract projections like "Future 100-year droughts will be n days longer," which can fail to provide concrete insights into the specifics of what a 100-year drought might entail even in the current climate, storylines offer more actionable and tangible scenarios. While the analysis has primarily aimed to enhance the understanding of future severe drought conditions at the provincial scale, HYDROTEL's distributed nature allows for the tailoring of hydrological indicators to address local-scale issues. In preparation for a*

*stakeholder workshop planned for the second phase of the project, additional local data is being gathered from nearly 20 participants representing governmental, municipal, industrial, health, environmental, and recreotourism sectors within a watershed north of the St. Lawrence River. By integrating their input with the modeled future events developed in this study, a potential storyline might read as follows: "A drought similar to the one experienced in 2021, but occurring under a +3°C global temperature increase, could persist for up to 26 additional days and lead to a further 30–50% reduction in monthly streamflow. This would have severe environmental and socioeconomic consequences. Streamflow would remain below the environmental flow threshold for over 90 days — more than double the 2021 duration — indicating a potentially catastrophic decline in water quality and biodiversity. Compounding the crisis, between August and October, the nearby city's water demand could consume up to 75% of the river's flow, not accounting for future consumption increases. Recreational tourism, a key economic activity in the region, would also suffer: kayaking restrictions due to insufficient water levels, which lasted 7 days in 2021, could extend up to two months under such extreme conditions." This event-based, context-rich approach allows stakeholders to recognize vulnerabilities by anchoring them to a familiar reference point — the 2021 drought — and projecting plausible future impacts. The city mentioned above was not concerned about water shortages in 2021, but understanding that a future event could last a month longer and leave barely enough water to meet even current needs may prompt proactive adaptation planning, such as bolstering water conservation measures, investing in infrastructure improvements, exploring alternative water sourcing strategies, or developing protocols to avoid conflicts between various users. Similarly, confronting certain water-dependent industries in that watershed with the possibility of future acute water shortages, regardless of existing water withdrawal agreements, could prove eye-opening and reveal potential vulnerabilities in their profitability or operational resilience.*